# QUANTIZATION BOUNDS FOR WASSERSTEIN METRICS

## ABSTRACT

The Wasserstein metric is becoming increasingly important in many machine learning applications such as generative modeling, image retrieval and domain adaptation. Despite its appeal, it is often too costly to compute. This has motivated approximation methods like entropy-regularized optimal transport, downsampling, and subsampling, which trade accuracy for computational efficiency. In this paper, we consider the challenge of computing efficient approximations to the Wasserstein metric that also serve as strict upper or lower bounds, as these are essential components of branch-and-bound, A$^*$ path finding, and heuristic search techniques in tasks such as trajectory inference, alignment, and clustering. Focusing on discrete measures on regular grids, our approach involves formulating and exactly solving a Kantorovich problem on a coarse grid using a quantized measure with a tailored cost matrix, followed by an upscaling and correction stage. This is done either in the primal or dual space to obtain valid upper and lower bounds on the Wasserstein metric of the full-resolution inputs. We evaluate our methods on the DOTmark optimal transport images benchmark as well as alignment tasks on volumetric dataset of macromolecules, demonstrating a 10×–100× speedup compared to entropy-regularized OT while keeping the approximation error well below 5% in 2D, and 30% width bounding regions in 3D.

## 1 INTRODUCTION

The Wasserstein metric is a basic tool in machine learning with broad applications in fields as diverse as computer vision, natural language processing, domain adaptation, and computational biology (Arjovsky et al., 2017; Kusner et al., 2015; Courty et al., 2017; Schiebinger et al., 2019). However, a limiting factor to its adoption is computational cost. For example, consider the calculation of the Wasserstein metric between $N \times N$ images that we treat as discrete measures on a regular grid. Each image has $N^2$ pixels, so computing the Wasserstein metric involves solving a linear program with $N^2 \times N^2$ variables and $\Theta(N^2)$ constraints. This is typically done using a network simplex algorithm whose worst-case runtime in this case is $O(N^6 \log N)$ (Peyré & Cuturi, 2019). For 3D signals, the computational cost is even worse at $O(N^9 \log N)$.

Our work is motivated by recent applications that use Wasserstein metrics as targets for minimization in diverse tasks such as trajectory inference, alignment, and clustering (Schiebinger et al., 2019; Banerjee et al., 2025; Riahi et al., 2023; Singer & Yang, 2024; Rao et al., 2020; Papayiannis et al., 2021). These optimization problems could be solved more efficiently using branch-and-bound, A$^*$ path finding, or other heuristic search techniques. However, this requires fast methods to compute *bounds* on the exact Wasserstein metric. Motivated by this challenge, the goal of this work is to develop fast approximations to the $p$-Wasserstein metric that also bound it from above or from below. Our particular focus is on image and volumetric data, but the approach could be generalized to other domains such as point clouds and graphs.

**Our contribution** We propose four methods for computing fast bounds on the Wasserstein metric between measures defined on regular grids: *weighted-cost upper bound*, *min-cost lower bound*, *primal upscaling upper bound*, and *dual upscaling lower bound*. Each method solves an exact optimal transport problem on a quantized (lower resolution) grid with a tailored ground cost and lifts the result to the original resolution. As a baseline for comparisons, we explain how entropy-regularized optimal transport can be used to produce bounds. See Section 3 and the supplementary for the description of the methods and Section 4 for a complexity analysis. For all methods, we developed efficient

JAX-based GPU implementations which we release as free software. To compare the accuracy and running time of the methods, we tested them on 2D images from the DOTmark dataset and on 3D volumetric data from the electron microscopy data bank (EMDB). The datasets and benchmark results are presented in Section 5 and in the appendix.

### 1.1 RELATED WORK

In recent years, many authors have developed fast approximations to the Wasserstein metric. A non-exhaustive list includes entropy-regularized optimal transport (Cuturi, 2013; Altschuler et al., 2018), convolutional Wasserstein (Solomon et al., 2015), sliced-Wasserstein (Deshpande et al., 2018; Kolouri et al., 2019), and linear approximations (Shirdhonkar & Jacobs, 2008; Moosmüller & Cloninger, 2023; Craig & Yu, 2025). The quantization-based approach of Beugnot et al. (2021) is similar in spirit to ours, but is focused on continuous measures and only guarantees the quality of approximation in expectation. Other methods that are related to our approach use hierarchical refinement to obtain a solution to the Kontorovich problem (Mérigot, 2011; Gerber & Maggioni, 2017; Feydy et al., 2021; Chen et al., 2022). However, unlike our approach, these methods do not give bounds on the exact Wasserstein metric.

Some works dedicated to accelerating similarity search and retrieval tasks, derive computationally efficient lower bounds for the Wasserstein metric. Yang et al. (2022) discusses several linear-time lower bounds for pruning search spaces, providing loose bounds with no tightness guarantees. Tree-based embeddings like Indyk & Thaper (2003); Backurs et al. (2020) provide bounds by embedding the metric into hierarchical structures, though at the cost of multiplicative $O(\log n)$ approximation factor. Another class of bounds arises from relaxing the transport constraints, as seen in the *relaxed word mover's distance* (Kusner et al., 2015; Atasu & Mittelholzer, 2019). While all these methods successfully provide valid lower bounds essential for large-scale retrieval, they prioritize computational speed over the tightness of the bounds. See Montesuma et al. (2025) for a recent survey of computation and approximation methods.

## 2 BACKGROUND

**Notation** We denote the non-negative real numbers by $\mathbb{R}_+$ and the set of integers $\{1, \ldots, n\}$ by $[n]$. The tensor product is denoted by $\otimes$ whereas pointwise multiplication and division are denoted by $\odot$ and $\oslash$, respectively. The $L^p$ norm of a vector is $\|\cdot\|_p$. The all-ones column vector is $\mathbf{1}_n \in \mathbb{R}^n$. The standard vector inner product is denoted by $\langle \cdot, \cdot \rangle$ and we use the same notation for the inner products of matrices. The support of a matrix $A \in \mathbb{R}^{n \times m}$ is the set of indices of nonzero elements $\operatorname{supp} A = \{(i, j) \in [n] \times [m] \mid A_{i,j} \neq 0\}$. The cardinality of a set $S$ is denoted by $\#S$. $\delta_x$ is the Dirac delta function at point $x$. Complete list of notations used in the paper is provided in Table 7.

### 2.1 OPTIMAL TRANSPORT

In the following, we give a quick review of basic concepts from optimal transport and refer the reader to Peyré & Cuturi (2019) for a more thorough introduction. Consider two discrete probability measures $\mu, \nu$ with point masses at $\mathcal{X} = \{x_1, \ldots, x_n\}$ and $\mathcal{Y} = \{y_1, \ldots, y_m\}$ respectively. We can express the measures as a sum of Dirac delta functions,

$$\mu = \sum_{i=1}^{n} \mu_i \delta_{x_i}, \qquad \nu = \sum_{j=1}^{m} \nu_j \delta_{y_j} . \tag{1}$$

We identify the measures with their non-negative coefficient vectors $\boldsymbol{\mu} \in \mathbb{R}_+^n, \boldsymbol{\nu} \in \mathbb{R}_+^m$. Since $\mu$ and $\nu$ are probability measures, we must have $\boldsymbol{\mu} \in \Sigma_n$ and $\boldsymbol{\nu} \in \Sigma_m$ where

$$\Sigma_N := \left\{ (p_1, \ldots, p_N) \in \mathbb{R}_+^N : p_1 + \cdots + p_N = 1 \right\} \tag{2}$$

is the probability simplex. The set of *coupling* matrices between $\mu$ and $\nu$ is

$$\Pi(\boldsymbol{\mu}, \boldsymbol{\nu}) := \left\{ \pi \in \mathbb{R}_+^{n \times m} \big| \pi \mathbf{1}_m = \boldsymbol{\mu}, \ \pi^\top \mathbf{1}_n = \boldsymbol{\nu} \right\} . \tag{3}$$

Each coupling can be viewed as a transport plan between $\mu$ and $\nu$ where $\pi_{ij}$ is the amount of mass transported from $x_i$ to $y_j$. The marginal constraints $\pi \mathbf{1}_m = \boldsymbol{\mu}$ mean that the entire source measure

$\mu$ is transported, whereas the constraints $\pi^{\top}\mathbf{1}_n = \boldsymbol{\nu}$ mean that this transport results in the target measure $\nu$. In particular, the set $\Pi(\boldsymbol{\mu}, \boldsymbol{\nu})$ always contains the trivial coupling $\pi_{\otimes} := \boldsymbol{\mu} \otimes \boldsymbol{\nu}$ that distributes every point mass in $\mathcal{X}$ to all point masses in $\mathcal{Y}$ proportionately to $\boldsymbol{\nu}$. Let $C \in \mathbb{R}_+^{n \times m}$ be a ground-cost matrix, where $C_{ij}$ represents the cost of transporting a unit mass from $x_i \in \mathcal{X}$ to $y_j \in \mathcal{Y}$. The Kantorovich problem is the minimization problem that seeks a cost-minimizing coupling between $\mu$ and $\nu$,

$$L_C(\mu, \nu) := \min_{\pi \in \Pi(\boldsymbol{\mu}, \boldsymbol{\nu})} \langle \pi, C \rangle. \tag{4}$$

This is a linear optimization problem with linear constraints. It admits a dual program,

$$L_C(\mu, \nu) = \max_{(\boldsymbol{f}, \boldsymbol{g}) \in \mathcal{R}(C)} \langle \boldsymbol{f}, \boldsymbol{\mu} \rangle + \langle \boldsymbol{g}, \boldsymbol{\nu} \rangle, \tag{5}$$

where

$$\mathcal{R}(C) := \{(\boldsymbol{f}, \boldsymbol{g}) \in \mathbb{R}^n \times \mathbb{R}^m | \forall i, j : f_i + g_j \leq C_{ij}\} \tag{6}$$

is the set of admissible dual potentials, also known as Kantorovich potentials. Given a distance metric $\rho : \mathcal{X} \times \mathcal{X} \to \mathbb{R}_+$, for any $p \geq 1$ the *Wasserstein-p* metric is a metric over the space of discrete probability measures with point masses at $\mathcal{X} = \{x_1, \ldots, x_n\}$, defined as $\mathcal{W}_p(\mu, \nu) := L_C(\mu, \nu)^{\frac{1}{p}}$ where $C_{ij} = \rho(x_i, y_j)^p$.

## 2.2 Measure coarsening

Consider a $d$-dimensional regular grid $\mathcal{X} = [N]^d$ and suppose that $N = n\kappa$ for some integers $n, \kappa$. One may subdivide the grid along the axes into a set $\mathbb{X}(\kappa) := \{X_1, \ldots, X_{n^d}\}$ of non-overlapping hypercubes of cardinality $\kappa^d$ that cover the entire grid $\mathcal{X}$, such that $N = n\kappa$. We define the coarse grid $\tilde{\mathcal{X}} := \{\tilde{x}_1, \ldots, \tilde{x}_{n^d}\}$ as the set of all hypercube centers, with $\tilde{x}_k = \texttt{mean}(X_k)$. Given a discrete measure $\mu$ over the grid $\mathcal{X}$, it induces a coarsened discrete measure by aggregating the mass of each hypercube and placing it at its center point. This coarsening corresponds to the $\texttt{SumPool}$ and $\texttt{AvgPool}$ operations on the measure and the coordinates, respectively, with size and stride $\kappa$

$$\texttt{SumPool}(\mu; \mathbb{X}(\kappa))_k := \sum_{x \in X_k} \mu(x) \qquad \texttt{AvgPool}(\mathcal{X}; \mathbb{X}(\kappa))_k := \frac{1}{\#X_k} \sum_{x \in X_k} x. \tag{7}$$

# 3 Methods

In this section, we describe several algorithms for computing bounds of the Wasserstein distance $\mathcal{W}_p(\mu, \nu)$ on a regular grid. First, in Section 3.1 we explain how entropic optimal transport can be used to obtain strict bounds. In the following subsections we introduce several novel algorithms for fast bounds on the Wasserstein metric that are based on quantization (or downscaling) of the inputs onto a coarse grid and sum-pooling the measures. These four methods have the following structure: First, they build a particular cost matrix for the coarse grid that takes the original measures into account. This is followed by a correction stage that upscales the solution on the coarse grid to a solution on the original grid and corrects the marginals using iterative proportional-fitting. The upscaling and correction stage is done separately in the primal and dual spaces, to obtain upper and lower bounds (respectively). Finally, to guarantee the correctness of the bounds without relying on the convergence of the proportional fitting procedure, we introduce an additional total variation correction term. The four quantization-based bounds, assuming the ground cost is Lipschitz continuous, are equal to the exact Wasserstein metric up to an additive $O(\kappa\sqrt{d})$ term, as detailed in Appendix D.

## 3.1 Bounds based on Entropy regularized OT

Entropy regularized optimal transport, also known as Sinkhorn distance (Cuturi, 2013), adds an entropy term $\mathcal{H}(\pi) = -\sum_{x,y} \pi(x, y) \log \pi(x, y)$ to the primal:

$$L_C^{\varepsilon}(\mu, \nu) := \min_{\pi \in \Pi(\boldsymbol{\mu}, \boldsymbol{\nu})} \langle \pi, C \rangle - \varepsilon \mathcal{H}(\pi). \tag{8}$$

This makes the problem strongly convex and solvable using Sinkhorn iterations (Knopp & Sinkhorn, 1967). In this section we will show how computing entropy regularized OT can be used to construct upper and lower bounds on the exact Wasserstein distance.

**Lower bound**  The dual form of the Sinkhorn distance is

$$L_C^\varepsilon(\mu, \nu) := \max_{\boldsymbol{f} \in \mathbb{R}^n, \boldsymbol{g} \in \mathbb{R}^m} \langle \boldsymbol{f}, \boldsymbol{\mu} \rangle + \langle \boldsymbol{g}, \boldsymbol{\nu} \rangle - \varepsilon \langle e^{\boldsymbol{f}/\varepsilon}, K e^{\boldsymbol{g}/\varepsilon} \rangle \tag{9}$$

where $K_{ij} := e^{-C_{ij}/\varepsilon}$ is the Gibbs kernel. The algorithmic solution, defined by the use of a finite number of iterations $t$ that achieves some stopping criteria, is known to satisfy a lower bound. Summarized in the following proposition:

**Proposition 3.1.** *Let $\hat{\boldsymbol{f}}_\varepsilon^{(t)}, \hat{\boldsymbol{g}}_\varepsilon^{(t)}$ be the iterations of the Sinkhorn distance algorithm in step $t \in \mathbb{N}$.*

$$\langle \hat{\boldsymbol{f}}_\varepsilon^{(t)}, \boldsymbol{\mu} \rangle + \langle \hat{\boldsymbol{g}}_\varepsilon^{(t)}, \boldsymbol{\nu} \rangle \leq L_C(\mu, \nu) \tag{10}$$

*as soon as $t \geq 1$.*

This follows directly from Peyré & Cuturi (2019, Propositions 4.5,4.8), so we can define $\underline{\mathcal{W}}_{p,\varepsilon}(\mu, \nu) := (\langle \hat{\boldsymbol{f}}_\varepsilon^{(t)}, \boldsymbol{\mu} \rangle + \langle \hat{\boldsymbol{g}}_\varepsilon^{(t)}, \boldsymbol{\nu} \rangle)^{\frac{1}{p}} \leq \mathcal{W}_p(\mu, \nu)$ for all $p \geq 1$.

**Upper bound**  Consider $d$-dimensional regular grids with side length $N \in \mathbb{N}$, $\mathcal{X} = \mathcal{Y} = [N]^d$, with discrete measures $\boldsymbol{\mu}, \boldsymbol{\nu} \in \Sigma_{N^d}$. Although the converged regularized optimal coupling

$$\pi_\varepsilon^* = \underset{\pi \in \Pi(\boldsymbol{\mu}, \boldsymbol{\nu})}{\arg \min} \langle \pi, C \rangle - \varepsilon \mathcal{H}(\pi) \tag{11}$$

defines an upper bound on the optimal transport $\langle \pi_\varepsilon^*, C \rangle \geq L_C(\mu, \nu)$, the algorithmic solution $\hat{\pi}_\varepsilon^{(t)}$ does not, since the marginals $\hat{\boldsymbol{\mu}}_\varepsilon^{(t)} = \hat{\pi}_\varepsilon^{(t)} \mathbf{1}_N$, $\hat{\boldsymbol{\nu}}_\varepsilon^{(t)} = (\hat{\pi}_\varepsilon^{(t)})^\top \mathbf{1}_N$ do not identify with the couplings $\boldsymbol{\mu}, \boldsymbol{\nu}$. We bound the effect of this difference using the weighted total variation. For some $x_0 \in \mathcal{X}$, using distance weights $\boldsymbol{w} = \{\rho(x_0, x)^p\}_{x \in \mathcal{X}}$ the Wasserstein-$p$ distance is controlled by *weighted total variation* (TV) (Villani, 2009),

$$\mathcal{TV}_p^{\boldsymbol{w}}(\mu, \nu) := 2^{1-\frac{1}{p}} \langle \boldsymbol{w}, |\boldsymbol{\mu} - \boldsymbol{\nu}| \rangle^{\frac{1}{p}} \geq \mathcal{W}_p(\mu, \nu). \tag{12}$$

Here $|\cdot|$ is the element-wise absolute value. We define the TV-corrected upper bound by

$$\overline{\mathcal{W}}_{p,\varepsilon}(\mu, \nu) := \langle \hat{\pi}_\varepsilon^{(t)}, C \rangle^{\frac{1}{p}} + \Delta_{\hat{\mu}_\varepsilon^{(t)}} + \Delta_{\hat{\nu}_\varepsilon^{(t)}}, \tag{13}$$

where $\Delta_{\hat{\mu}_\varepsilon^{(t)}} = \mathcal{TV}_p^{\boldsymbol{w}}(\hat{\mu}_\varepsilon^{(t)}, \mu)$, $\Delta_{\hat{\nu}_\varepsilon^{(t)}} = \mathcal{TV}_p^{\boldsymbol{w}}(\nu, \hat{\nu}_\varepsilon^{(t)})$ are the marginal corrections with weights $\boldsymbol{w} = \{\rho(\bar{x}, x_i)\}_i$ taken from the center of the measure $\bar{x} = \texttt{mean}(\mathcal{X})$. Using the triangle inequality for the Wasserstein metric, we can write

**Lemma 3.2.** *Let $\mu, \hat{\mu}, \nu, \hat{\nu} \in \Sigma_{N^d}$ discrete measures on $\mathcal{X} = [N]^d$, and $\hat{\pi} \in \Pi(\hat{\mu}, \hat{\nu})$ is a coupling between $\hat{\mu}$ and $\hat{\nu}$. For $p \geq 1$, $\mathcal{W}_p(\mu, \nu) \leq \langle \hat{\pi}, C \rangle^{\frac{1}{p}} + \Delta_{\hat{\mu}} + \Delta_{\hat{\nu}}$.*

Combining this with (13) shows that $\mathcal{W}_p(\mu, \nu) \leq \overline{\mathcal{W}}_{p,\varepsilon}(\mu, \nu)$. See illustration in Figure 1.

**Proposition 3.3.** *Let $\mu, \hat{\mu}, \nu, \hat{\nu} \in \Sigma_n$ be discrete measures in $\mathcal{X}$ and $\xi > 0$, satisfying the convergence criteria $\|\hat{\boldsymbol{\mu}} - \boldsymbol{\mu}\|_1 + \|\boldsymbol{\nu} - \hat{\boldsymbol{\nu}}\|_1 < \xi$. The sum of the marginal corrections is bounded,*

$$\Delta_{\hat{\mu}} + \Delta_{\hat{\nu}} < 2^{2-\frac{2}{p}} \xi^{\frac{1}{p}} r \tag{14}$$

*where $r := \max_{x \in \mathcal{X}} \{\rho(\bar{x}, x)\}$ is radius of $\mathcal{X}$.*

Proofs for Lemma 3.2 and Proposition 3.3 are provided in Appendix C.1.

### 3.2 WEIGHTED-COST UPPER BOUND

In this subsection we consider an approach to bound Wasserstein distance by downscaling the grid $\mathcal{X}$ to $\tilde{\mathcal{X}} = \tilde{\mathcal{Y}}$ and the measures to $\tilde{\mu}, \tilde{\nu} \in \Sigma_{n^d}$ using regular hypercubes as described in Section 2.2. We define the marginally weighted coarse cost

$$\bar{C}_{k\ell} := \frac{1}{\mu(X_k)\nu(Y_\ell)} \sum_{x \in X_k, y \in Y_\ell} \rho(x, y)^p \mu(x)\nu(y). \tag{15}$$

It follows that $\bar{C} = \texttt{SumPool}(C \odot \pi_\otimes; \mathbb{X}(\kappa)^2) \oslash \texttt{SumPool}(\pi_\otimes; \mathbb{X}(\kappa)^2)$, which can be used efficiently for small enough grids. Then we compute the optimal coupling for the marginally weighted coarse cost using network simplex solver (Bonneel et al., 2011), defining an upper bound

$$\overline{\mathcal{W}}_p^\otimes := L_{\bar{C}}(\tilde{\mu}, \tilde{\nu})^{\frac{1}{p}}. \tag{16}$$

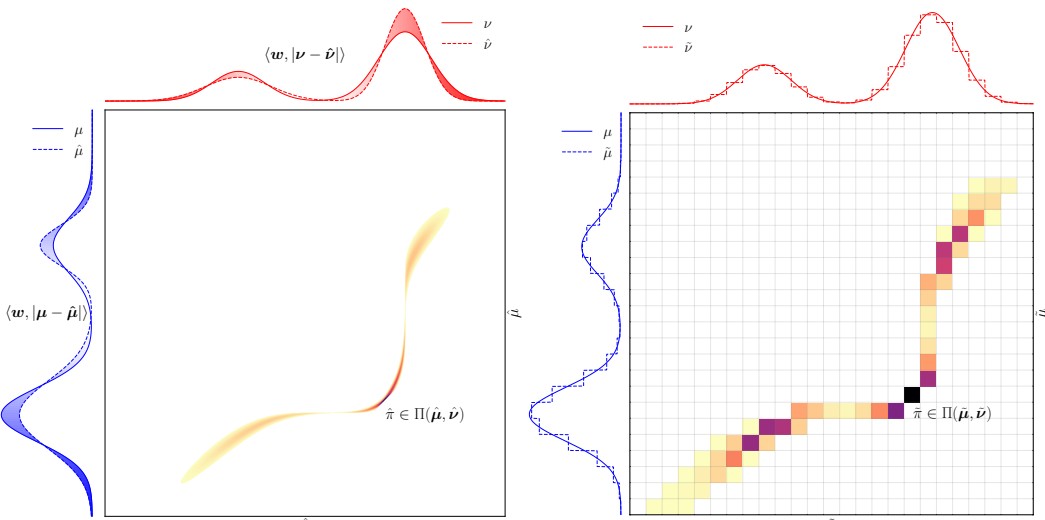

Figure 1: Optimal transport bounds visualization. Left: Illustration of optimal transport between discrete probability measures, approximating the marginal measures. Right: Illustration of optimal transport between quantized discrete probability measures, as approximation.

**Theorem 3.4.** *The optimal transport loss under the marginally weighted coarse cost $L_{\bar{C}}(\tilde{\mu}, \tilde{\nu})$ is an upper bound to the optimal transport loss $L_C(\mu, \nu)$, and similarly for the Wasserstein distance,*

$$L_{\bar{C}}(\tilde{\mu}, \tilde{\nu})^{\frac{1}{p}} \geq \mathcal{W}_p(\mu, \nu). \tag{17}$$

The proof uses an auxiliary coupling transferring the mass between each pair of sub-regions $(X_i, Y_j)$ using the trivial coupling, weighted by a coarse coupling, choosing the optimal coarse coupling. A detailed proof is provided in Appendix C.2.

### 3.3 MIN-COST LOWER BOUND

For the same coarsening, we define the locally minimal cost $C_{k\ell}^{\min} := \min_{x \in X_k, y \in Y_\ell} \rho(x, y)^p$, and compute the optimal coupling for this coarse cost using a network simplex solver, yielding a lower bound $\underline{\mathcal{W}}_p^{\min} := L_{C^{\min}}(\tilde{\mu}, \tilde{\nu})^{\frac{1}{p}}$. The following theorem is proved in Appendix C.3.

**Theorem 3.5.** *The coarse optimal transport cost set by the locally minimal cost, is a lower bound of the optimal transport. $L_{C^{\min}}(\tilde{\mu}, \tilde{\nu}) \leq L_C(\mu, \nu)$.*

### 3.4 PRIMAL UPSCALING UPPER BOUND

In this approach we upscale an optimal coupling for the coarse cost $\tilde{c}_{k\ell} = \rho(\tilde{x}_k, \tilde{y}_\ell)^p$ computed using a network simplex solver $\tilde{\pi}^* = \arg\min_{\tilde{\pi} \in \Pi(\tilde{\mu}, \tilde{\nu})} \langle \tilde{\pi}, \tilde{C} \rangle$ back to the original problem size.

**Up-scaled coupling** A coupling matrix $\pi$ of dimensions $n^d \times n^d$ can equivalently be represented as a $2d$-tensor of the shape $n \times n \times \cdots \times n$. We formally define operations for reshaping matrices into tensors and back. Let `reshape` be a cardinality-preserving transformation from $\mathbf{A}$ to $\mathbf{B}$ such that $a_{ij} = b_{u_1, \ldots, u_d, v_1, \ldots, v_d}$ where $(u_1, \ldots, u_d)$ is a multi-index that corresponds to a matrix row by $i = 1 + \sum_{k=1}^{d} u_k n^{k-1}$. Similarly the multi-index $(v_1, \ldots, v_d)$ corresponds to the column index $j$. Reshaping the coarse optimal coupling $\tilde{\pi}^*$ into a $2d$-tensor $\tilde{\mathbf{P}}^*$, we up-scale the optimal coupling using a normalized positive-valued kernel $\mathbf{K}$, a $2d$-tensor representing a hypercube of width $\kappa$. $\hat{\mathbf{P}} := \tilde{\mathbf{P}}^* \otimes \mathbf{K}$. By reshaping the up-scaled tensor $\hat{\mathbf{P}}$ into an up-scaled matrix $\hat{\pi}$, we obtain the approximate up-scaled coupling. Using a uniform kernel $\mathbf{K}$ performs nearest-neighbor interpolation.

**Lemma 3.6.** *Using a normalized positive-valued kernel* $\mathbf{K}$, *satisfying* $\sum_{t \in [\kappa]^{2d}} \mathbf{K}_t = 1$, *ensures* $\hat{\pi}$ *represents a coupling, thus satisfying* $\hat{\pi} \in \mathbb{R}_+^{N^d \times N^d}$ *and* $\sum_{i=1}^{N^d} \sum_{j=1}^{N^d} \hat{\pi}_{ij} = 1$, *such that* $\hat{\pi} \in \Pi(\hat{\pi}\mathbf{1}_{N^d}, \hat{\pi}^\top \mathbf{1}_{N^d})$.

**Iterative proportional fitting**  The approximate up-scaled coupling is $\xi$-fitted into $\hat{\pi}_\xi = \mathrm{diag}(\boldsymbol{a})\hat{\pi}\,\mathrm{diag}(\boldsymbol{b})$ by iterative proportional fitting using Sinkhorn–Knopp algorithm, until the marginals $\hat{\boldsymbol{\mu}}_\xi = \boldsymbol{a} \odot (\hat{\pi}\boldsymbol{b})$, $\hat{\boldsymbol{\nu}}_\xi = \boldsymbol{b} \odot (\hat{\pi}^\top \boldsymbol{a})$ are converged to $\|\hat{\boldsymbol{\mu}}_\xi - \boldsymbol{\mu}\|_1 + \|\boldsymbol{\nu} - \hat{\boldsymbol{\nu}}_\xi\|_1 < \xi$, where $\boldsymbol{a}, \boldsymbol{b} \in \mathbb{R}_+^{N^d}$ are the vector scale factors, yielding the approximation $\widehat{\mathcal{W}}_p(\mu, \nu) := \langle \hat{\pi}_\xi, C \rangle^{\frac{1}{p}}$.

**Upper bound**  Using weighted total variation (Equation (12)) we define an upscaling upper bound

$$\overline{\mathcal{W}}_p(\mu, \nu) := \widehat{\mathcal{W}}_p(\mu, \nu) + \Delta_{\hat{\mu}_\xi} + \Delta_{\hat{\nu}_\xi}, \tag{18}$$

**Theorem 3.7** (Upscaling Upper Bound). $\overline{\mathcal{W}}_p(\mu, \nu)$ *is upper bound of the Wasserstein distance,*

$$\mathcal{W}_p(\mu, \nu) \leq \overline{\mathcal{W}}_p(\mu, \nu). \tag{19}$$

*Proof of Theorem 3.7.* The up-scaled matrix $\hat{\pi} \in \Pi(\hat{\pi}\mathbf{1}_{N^d}, \hat{\pi}^\top \mathbf{1}_{N^d})$ is normalized as a coupling, by Lemma 3.6, and $\hat{\pi}_\xi$ retains this normalization, by Sinkhorn's theorem. $\overline{\mathcal{W}}_p(\mu, \nu)$ is an upper bound of the Wasserstein distance, by Lemma 3.2. $\qquad\square$

*Remark* 3.8. Considering $\mathcal{X} = [N]^d$ with $L^2$ norm, the radius becomes $r = \frac{1}{2}d^{\frac{1}{2}}N$ (see Proposition 3.3) such that the weighted total variation correction $\Delta_{\hat{\mu}} + \Delta_{\hat{\nu}}$ is at most $2^{1-\frac{2}{p}}d^{\frac{1}{2}}N\xi^{\frac{1}{p}}$. Thus, the weighted total variation correction is negligible for $\xi \ll N^{-p}$, and can be ignored for many practical use cases.

### 3.5   DUAL UPSCALING LOWER BOUND

We construct a lower bound for the Wasserstein distance $\mathcal{W}_p(\mu, \nu)$ by solving a down-scaled optimal transport problem using coarsened measures. The coarse optimal Kantorovich potentials are then up-scaled using a multi-linear interpolation and improved using a c-transform. Considering the same setting as in Section 3.4, we solve for the optimal potentials of the down-scaled discrete measures

$$(\tilde{\boldsymbol{f}}^*, \tilde{\boldsymbol{g}}^*) = \operatorname*{arg\,max}_{(\tilde{\boldsymbol{f}}, \tilde{\boldsymbol{g}}) \in \mathcal{R}(\tilde{C})} \langle \tilde{\boldsymbol{f}}, \tilde{\boldsymbol{\mu}} \rangle + \langle \tilde{\boldsymbol{g}}, \tilde{\boldsymbol{\nu}} \rangle \tag{20}$$

and evaluate the dual transport cost at the original scale by upscaling the optimal potentials. Upscaling can be performed by any multivariate interpolation method such as nearest-neighbor, spline, multi-linear and polynomial methods. Using an interpolation function $R : \mathcal{X} \cup \tilde{\mathcal{X}} \to \mathbb{R}$, the up-scaled potential $\hat{\boldsymbol{f}}$ is defined by

$$\hat{\boldsymbol{f}} := \{R_{\tilde{\boldsymbol{f}}, \tilde{\mathcal{X}}}(x_i)\}_{i \in [N^d]}. \tag{21}$$

An important property of the dual formulation is that for every potential $\boldsymbol{f} \in \mathbb{R}^n$ we can easily find a *tight* potential $\boldsymbol{f}^{\mathsf{c}} \in \mathbb{R}^m$ such that $(\boldsymbol{f}, \boldsymbol{f}^{\mathsf{c}}) \in \mathcal{R}(C)$, by $f_j^{\mathsf{c}} := \min_i C_{ij} - f_i$. This is known as a *c-transform*. It can be shown that repeating this process once more achieves a tight pair $(\boldsymbol{f}^{\mathsf{c}}, \boldsymbol{f}^{\mathsf{cc}}) \in \mathcal{R}(C)$, where $f_i^{\mathsf{cc}} := \min_j C_{ij} - f_j^{\mathsf{c}}$. Thus, a lower bound is guaranteed by using the c-transform to generate the potential pair from the up-scaled potential $\boldsymbol{f} = \hat{\boldsymbol{f}}^{\mathsf{cc}}$ and $\boldsymbol{g} = \hat{\boldsymbol{f}}^{\mathsf{c}}$, which yields the upscaling lower bound

$$\underline{\mathcal{W}}_p(\mu, \nu) := (\langle \boldsymbol{f}, \boldsymbol{\mu} \rangle + \langle \boldsymbol{g}, \boldsymbol{\nu} \rangle)^{\frac{1}{p}}. \tag{22}$$

Finally, by the admissibility of a c-transformed pair $(\boldsymbol{f}, \boldsymbol{g}) \in \mathcal{R}(C)$ we can write

**Proposition 3.9.** *Considering an approximate potential* $\hat{\boldsymbol{f}} \in \mathbb{R}^{N^d}$. *For* $\boldsymbol{f} = \hat{\boldsymbol{f}}^{\mathsf{cc}}$ *and* $\boldsymbol{g} = \hat{\boldsymbol{f}}^{\mathsf{c}}$

$$\langle \boldsymbol{f}, \boldsymbol{\mu} \rangle + \langle \boldsymbol{g}, \boldsymbol{\nu} \rangle \leq L_C(\mu, \nu). \tag{23}$$

## 4 COMPUTATIONAL COMPLEXITY ANALYSIS

The quantization-based bounds involve the following steps: computing the Wasserstein metric on the quantized measures, upscaling the dual potentials or couplings to the original scale, and the calculation of weighted total variation correction terms. The latter is calculated in linear time and space, thus negligible w.r.t. the other steps. In the following, we detail the computational gains provided by the proposed methods.

**Downscaled optimal transport** The solution to the Kantorovich problem of the scaled measures can be solved by dedicated linear programming methods, such as the network simplex used in Bonneel et al. (2011), with $O\left(n^{3d}\log n\right)$ time complexity (Ahuja et al., 1993). By solving only for the optimal transport of the coarse measures, we produce a computational speedup of $\Theta(\kappa^{3d})$ (up to log factors). The space complexity can also be significantly reduced, since one can avoid storing the full cost matrix of size $N^d \times N^d$, by using coarse cost matrices, e.g. $\bar{C}, C^{\min}$, and $\tilde{C}$ of size $n^d \times n^d$, realizing a memory gain of $\Theta(\kappa^{2d})$.

**Upscaled optimal coupling** The optimal coarse coupling $\tilde{\pi}^*$ is a sparse matrix with at most $2n^d - 1$ positive entries (Peyré & Cuturi, 2019, Proposition 3.4). Thus, the up-scaled approximate coupling from Theorem 3.7 conserves this sparsity with $\#\operatorname{supp}\hat{\pi} \le \kappa^{2d}(2n^d - 1)$, allowing to calculate the approximate optimal transport

$$\langle \hat{\pi}_\xi, C \rangle = \sum_{(i,j) \in \operatorname{supp}\hat{\pi}} \overbrace{a_i \hat{\pi}_{ij} b_j}^{(\hat{\pi}_\xi)_{ij}} \rho(x_i, y_j)^p. \tag{24}$$

without impacting the total time and space complexity of the coarse optimal transport solution.

$$\frac{\#C}{\#\operatorname{supp}\hat{\pi}} = \frac{N^{2d}}{\kappa^{2d}(2n^d - 1)} = \Theta\left((N/\kappa)^d\right). \tag{25}$$

**Lazy c-transform** To reduce the memory requirements of Equation (22), we evaluate the c-transform on-demand (i.e. "lazy") without storing the entire cost matrix $C$.

$$g_j \leftarrow \min_i \rho(x_i, y_j)^p - \hat{f}_i \tag{26}$$

$$f_i \leftarrow \min_j \rho(x_i, y_j)^p - g_j \tag{27}$$

| Method | Time Complexity | Space Complexity |
|---|---|---|
| Exact Wasserstein computation | $\tilde{O}\left(N^{3d}\right)$ | $O\left(N^{2d}\right)$ |
| Entropic Regularization-Based Bounds (Lin et al., 2022) | $\tilde{O}\left(N^{2d}/\varepsilon^2\right)$ | $O\left(N^{2d}\right)$ |
| Quantization-Based Bounds | $\tilde{O}\left((N/\kappa)^{3d}\right)$ | $O\left((N/\kappa)^{2d}\right)$ |

Table 1: Complexity of different bounds in terms of the fine-scale cardinality $\#\mathcal{X} = N^d = (n\kappa)^d$. The first row corresponds to the methods in Section 3.1 and the second to Sections 3.2 to 3.5.

While the entropic regularization-based bounds enjoy better asymptotic time complexity than the proposed quantization-based bounds, in practice, to achieve comparable accuracy we are required to pick very small $\varepsilon$ values, eliminating this effect, as shown in Section 5.

## 5 EXPERIMENTS

The methods were implemented in Python and optimized for GPU acceleration using the JAX numerical computing library (Bradbury et al., 2018). For solving entropy regularized optimal transport we used sinkhorn solver from OTT-JAX (Cuturi et al., 2022) and for solving exact Kontorovich problems we used the network-simplex emd solver from Python Optimal Transport (POT) (Flamary et al., 2021). The benchmarks were run on a machine with an AMD EPYC 9654 CPU and an NVIDIA L40 GPU.

**EMDB** In the field of structural biology, approximations of the Wasserstein metric are increasingly being used on 2D projection images and 3D volumetric reconstructions of proteins and other macromolecules. Specific applications include molecular alignment, clustering and dimensionality reduction, with most methods substituting the Wasserstein metric with a crude approximation that is fast to compute (Rao et al., 2020; Riahi et al., 2023; Singer & Yang, 2024; Kileel et al., 2021). We evaluate our algorithms in a challenging 3D alignment setting, where we wish to compute the Wasserstein-$p$ metric $p \in \{1, 2\}$ between rotated 3D density maps of the same molecule. The volumetric density maps are downloaded from the electron microscopy data bank (EMDB) (wwPDB Consortium, 2024) using the ASPIRE package (Wright et al., 2025). Figure 2 shows the computed bounds for rotations between $0°$ and $180°$ of the Plasmodium falciparum 80S ribosome 3D density map (Wong et al., 2014). Plots for other molecules are shown in the supplementary material. Dual upscaling (lower bound) and weighted-cost (upper bound) methods at $\kappa = 2$ provide the best approximations with average relative bounding region of only $31.37\% \pm 12.42\%$ ($p = 1$) and $28.06\% \pm 13.15\%$ ($p = 2$), where for each angle the relative bounding region is $\frac{|upper - lower|}{upper}$. A summary of the computational speed-up is provided in Table 2.

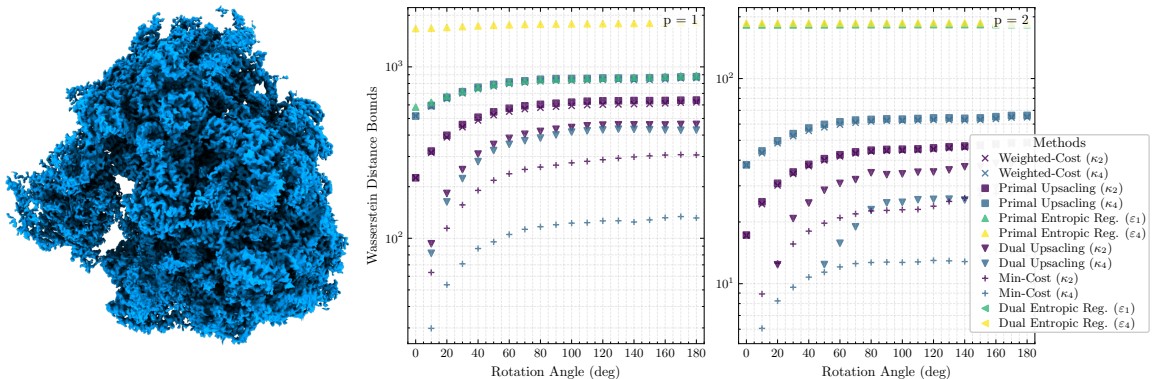

Figure 2: Wasserstein distance bounds between rotated 3D density maps of the 80S ribosome. The left panel shows an isosurface plot of the 3D density map that we rotated around the $z$-axis. The other two panels compare the different algorithms for producing upper and lower bounds on the Wasserstein-$p$ metric. Due to the logarithmic scale of the y-axis, the zero-valued dual entropic regularization bounds do not appear in the plot. (center) $p = 1$; (right) $p = 2$.

| | Upper Bounds | | | | | | Lower Bounds | | | | | |
|---|---|---|---|---|---|---|---|---|---|---|---|---|
| | Weighted-Cost | | Primal Upscaling | | Entropic Regularization | | Dual Upscaling | | Min-Cost | | Entropic Regularization | |
| p | $\kappa_2$ | $\kappa_4$ | $\kappa_2$ | $\kappa_4$ | $\varepsilon_1$ | $\varepsilon_4$ | $\kappa_2$ | $\kappa_4$ | $\kappa_2$ | $\kappa_4$ | $\varepsilon_1$ | $\varepsilon_4$ |
| 1 | 0.22% | **0.12%** | 1.09% | 27.23% | 130.95% | 144.97% | 0.55% | 0.33% | 0.20% | **0.11%** | 190.88% | 197.35% |
| | $\pm 0.06\%$ | $\pm 0.26\%$ | $\pm 0.34\%$ | $\pm 15.99\%$ | $\pm 29.46\%$ | $\pm 39.00\%$ | $\pm 0.21\%$ | $\pm 0.21\%$ | $\pm 0.06\%$ | $\pm 0.29\%$ | $\pm 58.51\%$ | $\pm 52.12\%$ |
| 2 | 0.22% | **0.06%** | 0.30% | 0.16% | 164.64% | 171.42% | 0.19% | **0.04%** | 0.21% | **0.04%** | 195.49% | 197.76% |
| | $\pm 0.04\%$ | $\pm 0.02\%$ | $\pm 0.07\%$ | $\pm 0.08\%$ | $\pm 42.18\%$ | $\pm 41.89\%$ | $\pm 0.05\%$ | $\pm 0.02\%$ | $\pm 0.05\%$ | $\pm 0.02\%$ | $\pm 42.43\%$ | $\pm 44.34\%$ |

Table 2: Computation time relative to the exact Wasserstein distance computation for 3D Cryo-EM data at $32 \times 32 \times 32$ resolution. Results show mean $\pm$ standard deviation across rotations.

**DOTmark** The methods were evaluated on 2D images from the discrete optimal transport benchmark (Schrieber et al., 2017), using $\rho = L^2$ the euclidean metric, at $p = \{1, 2\}$. To examine the effect of the scaling factor, the quantization-based methods were evaluated using $\kappa = \{2, 4\}$. To examine the effect of the entropic-regularization parameter, the regularization-based methods were evaluated using $\varepsilon = \{0.001N^p, 0.004N^p\}$ explicitly dependent on $N^p$ term to avoid large $\|C\|_\infty/\varepsilon$ causing numerical instability (Altschuler et al., 2018), since $\|C\|_\infty \propto N^p$ in our setting. For upper bounds, while at $p = 1$ the entropic-regularization upper bound at $\varepsilon = 0.001N^p$ delivers the best

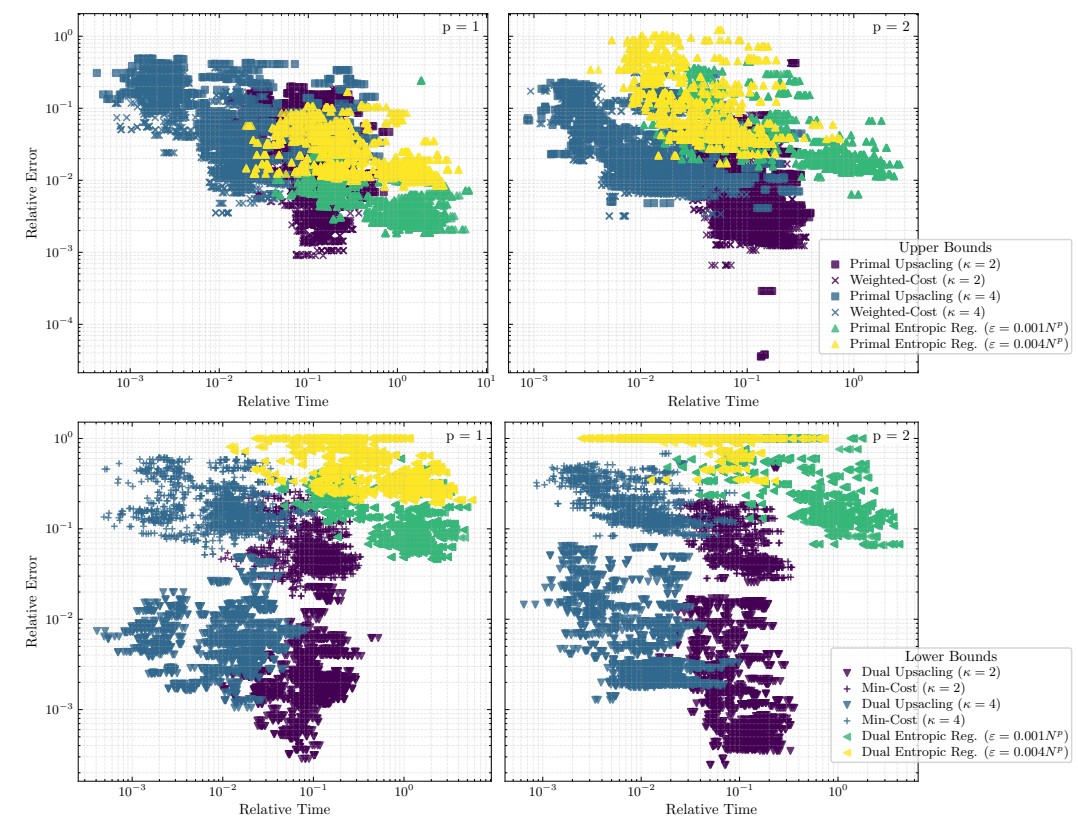

Figure 3: This figure shows the time/accuracy tradeoff of the different bounds on image pairs from the DOTMark dataset. Each point in the plots represents an image pair whose Wasserstein metric is estimated using a particular method. The $x$ value of the point is the time taken to compute the estimate, reltive to the runtime of the exact Wasserstein computation. The $y$ value is the relative error of the bound. (top left) upper bounds for $W_1$. (top right) upper bounds for $W_2$. (bottom left) lower bounds for $W_1$. (lower right) lower bounds for $W_2$.

approximation, summarized in Table 4 in Appendix B, it does so with significant impact on the computation time, as seen in Figure 3. In contrast, for both upper and lower bounds, the quantization methods yield the best approximations at $\kappa = 2$, maintaining low computation time and ranking second in the relative time benchmark after the quantization methods scaled at $\kappa = 4$.

## 6 CONCLUSION AND DISCUSSION

In this paper, we proposed several methods for computing fast approximations that lower or upper-bound the Wasserstein metric between discrete distributions on a regular grid. Our experiments on 2D images and 3D volumetric data demonstrate significant improvements in computational efficiency and accuracy compared to bounds based on entropic OT. In future work, our approach could be refined and extended by exploring different methods for the downscaling and upscaling stages in addition to multi-scale approaches. The methods could also be extended to domains beyond regular grids such as point clouds in $\mathbb{R}^n$ and graphs by picking coarsening and interpolation operations that are suitable for the domain.

**Code availability and reproducibility**    Python implementations of all the methods in the paper, as well as code for reproducing the figures and tables, will be released as free software on Github.

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

Lower-bound based on entropic regularization described in Section 3.1. This method simply runs the iterative Sinkhorn algorithm an then returns the unregularized cost term of the dual solution to the entropic regularization problem.

---

**Algorithm 1** Regularization-Based Lower Bound

---

**Require:** $\boldsymbol{\mu}, \boldsymbol{\nu} \in \Sigma_{N^d}$ on $\mathcal{X} = [N]^d$, $p \geq 1, \varepsilon > 0, \xi > 0$ and metric $\rho$.
$\quad C \leftarrow \{\rho(x_i, y_j)^p\}_{ij}$
$\quad K \leftarrow \exp\left(-\frac{C}{\varepsilon}\right)$
$\quad$ Initialize $\boldsymbol{f} \leftarrow \boldsymbol{0}_n, \boldsymbol{g} \leftarrow \boldsymbol{0}_m$
$\quad$ Initialize $\boldsymbol{b} \leftarrow \exp\left(\frac{\boldsymbol{g}}{\varepsilon}\right)$
$\quad$ **repeat**
$\quad\quad \boldsymbol{a} \leftarrow \boldsymbol{\mu} \oslash (K\boldsymbol{b})$
$\quad\quad \boldsymbol{f} \leftarrow \varepsilon \log \boldsymbol{a}$
$\quad\quad \boldsymbol{b} \leftarrow \boldsymbol{\nu} \oslash (K^\top \boldsymbol{a})$
$\quad\quad \boldsymbol{g} \leftarrow \varepsilon \log \boldsymbol{b}$
$\quad$ **until** $\|\boldsymbol{a} \odot K\boldsymbol{b} - \boldsymbol{\mu}\|_1 + \|\boldsymbol{\nu} - \boldsymbol{b} \odot K^\top \boldsymbol{a}\|_1 < \xi$
$\quad$ **return** $\left(\langle \boldsymbol{f}, \boldsymbol{\mu}\rangle + \langle \boldsymbol{g}, \boldsymbol{\nu}\rangle\right)^{\frac{1}{p}}$

---

Algorithm for entropic regularization-based upper bound described in Section 3.1, using the un-regularized term of the primal solution to the entropic regularization problem, with weighted total variation marginal corrections.

---

**Algorithm 2** Regularization-Based Upper Bound

---

**Require:** $\boldsymbol{\mu}, \boldsymbol{\nu} \in \Sigma_{N^d}$ on $\mathcal{X} = [N]^d$, $p \geq 1, \varepsilon > 0, \xi > 0$ and metric $\rho$.
$\quad C \leftarrow \{\rho(x_i, y_j)^p\}_{ij}$
$\quad K \leftarrow \exp\left(-\frac{C}{\varepsilon}\right)$
$\quad$ Initialize $\boldsymbol{b} \leftarrow \boldsymbol{1}_{N^d}$
$\quad$ **repeat**
$\quad\quad \boldsymbol{a} \leftarrow \boldsymbol{\mu} \oslash K\boldsymbol{b}$
$\quad\quad \boldsymbol{b} \leftarrow \boldsymbol{\nu} \oslash K^\top \boldsymbol{a}$
$\quad\quad \hat{\boldsymbol{\mu}} \leftarrow \boldsymbol{a} \odot (K\boldsymbol{b})$
$\quad\quad \hat{\boldsymbol{\nu}} \leftarrow \boldsymbol{b} \odot (K^\top \boldsymbol{a})$
$\quad$ **until** $\|\hat{\boldsymbol{\mu}} - \boldsymbol{\mu}\|_1 + \|\boldsymbol{\nu} - \hat{\boldsymbol{\nu}}\|_1 < \xi$
$\quad \hat{\pi}_\varepsilon \leftarrow \operatorname{diag}(\boldsymbol{a}) K \operatorname{diag}(\boldsymbol{b})$
$\quad \bar{x} \leftarrow \operatorname{mean}(\mathcal{X})$
$\quad \boldsymbol{w} \leftarrow \{\rho(\bar{x}, x_i)^p\}_i$
$\quad \Delta_{\hat{\mu}} \leftarrow 2^{1-\frac{1}{p}} \langle \boldsymbol{w}, |\hat{\boldsymbol{\mu}} - \boldsymbol{\mu}|\rangle^{\frac{1}{p}}$
$\quad \Delta_{\hat{\nu}} \leftarrow 2^{1-\frac{1}{p}} \langle \boldsymbol{w}, |\boldsymbol{\nu} - \hat{\boldsymbol{\nu}}|\rangle^{\frac{1}{p}}$
$\quad$ **return** $\langle \hat{\pi}_\varepsilon, C\rangle^{\frac{1}{p}} + \Delta_{\hat{\mu}} + \Delta_{\hat{\nu}}$

---

Algorithm for quantization-based upper bound described in Section 3.2, using coarse cost weighted by the trivial coupling.

**Algorithm 3** Weighted-Cost Upper Bound

---

**Require:** $\boldsymbol{\mu}, \boldsymbol{\nu} \in \Sigma_{N^d}$ on $\mathcal{X} = [N]^d$, $p \geq 1$, $\kappa \in \mathbb{N}$ and metric $\rho$.

$\tilde{\boldsymbol{\mu}} \leftarrow \texttt{SumPool}(\boldsymbol{\mu}; \kappa)$

$\tilde{\boldsymbol{\nu}} \leftarrow \texttt{SumPool}(\boldsymbol{\nu}; \kappa)$

$\bar{C}_{k\ell} \leftarrow \left\{ \frac{1}{\tilde{\mu}_k \tilde{\nu}_\ell} \sum_{\substack{x \in X_k \\ y \in Y_\ell}} \rho(x, y)^p \mu(x)\nu(y) \right\}_{k\ell}$

Solve $L_{\bar{C}} \leftarrow \min_{\tilde{\pi} \in \Pi(\tilde{\boldsymbol{\mu}}, \tilde{\boldsymbol{\nu}})} \langle \tilde{\pi}, \bar{C} \rangle$

**return** $L_{\bar{C}}^{\frac{1}{p}}$

---

Algorithm for bi-level quantization-based upper bound described in Section 3.4, using nearest-neighbor upscaling of the optimal coarse coupling, iterative proportional fitting of the marginals (i.e. Sinkhorn iterations), with weighted total variation marginal corrections.

**Algorithm 4** Upscaling Upper Bound

---

**Require:** $\boldsymbol{\mu}, \boldsymbol{\nu} \in \Sigma_{N^d}$ on $\mathcal{X} = [N]^d$, $p \geq 1$, $\kappa \in \mathbb{N}$, $\xi > 0$ and metric $\rho$.

$\tilde{\mathcal{X}} \leftarrow \texttt{AvgPool}(\mathcal{X}; \kappa)$

$\tilde{\boldsymbol{\mu}} \leftarrow \texttt{SumPool}(\boldsymbol{\mu}; \kappa)$

$\tilde{\boldsymbol{\nu}} \leftarrow \texttt{SumPool}(\boldsymbol{\nu}; \kappa)$

$\tilde{C} \leftarrow \{\rho(\tilde{x}_k, \tilde{x}_\ell)^p\}_{k\ell}$

Solve $\tilde{\pi}^* \leftarrow \arg\min_{\tilde{\pi} \in \Pi(\tilde{\boldsymbol{\mu}}, \tilde{\boldsymbol{\nu}})} \langle \tilde{\pi}, \tilde{C} \rangle$

▷ Up-scaled coupling

$\tilde{\mathbf{P}}^* \leftarrow \texttt{reshape}(\tilde{\pi}^*; n)$                      ▷ Reshape as tensor

$\mathbf{K} \leftarrow \{\kappa^{-2d}\}_{t \in [\kappa]^{2d}}$

$\hat{\mathbf{P}} \leftarrow \tilde{\mathbf{P}}^* \otimes \mathbf{K}$                              ▷ Upscaling

$\hat{\pi} \leftarrow \texttt{reshape}^{-1}(\hat{\mathbf{P}}; N)$

▷ Iterative proportional fitting

Initialize $\boldsymbol{b} \leftarrow \mathbf{1}_{N^d}$

**repeat**

    $\boldsymbol{a} \leftarrow \boldsymbol{\mu} \oslash \hat{\pi}\boldsymbol{b}$

    $\boldsymbol{b} \leftarrow \boldsymbol{\nu} \oslash \hat{\pi}^\top \boldsymbol{a}$

    $\hat{\boldsymbol{\mu}} \leftarrow \boldsymbol{a} \odot (\hat{\pi}\boldsymbol{b})$

    $\hat{\boldsymbol{\nu}} \leftarrow \boldsymbol{b} \odot (\hat{\pi}^\top \boldsymbol{a})$

**until** $\|\hat{\boldsymbol{\mu}} - \boldsymbol{\mu}\|_1 + \|\boldsymbol{\nu} - \hat{\boldsymbol{\nu}}\|_1 < \xi$

▷ Upper bound

$\widehat{\mathcal{W}}_p \leftarrow \left( \sum_{(i,j) \in \text{supp}(\hat{\pi})} a_i \hat{\pi}_{ij} b_j \, \rho(x_i, x_j)^p \right)^{\frac{1}{p}}$

$\bar{x} \leftarrow \texttt{mean}(\mathcal{X})$

$\boldsymbol{w} \leftarrow \{\rho(\bar{x}, x_i)^p\}_i$

$\Delta_{\hat{\mu}} \leftarrow 2^{1 - \frac{1}{p}} \langle \boldsymbol{w}, |\hat{\boldsymbol{\mu}} - \boldsymbol{\mu}| \rangle^{\frac{1}{p}}$

$\Delta_{\hat{\nu}} \leftarrow 2^{1 - \frac{1}{p}} \langle \boldsymbol{w}, |\boldsymbol{\nu} - \hat{\boldsymbol{\nu}}| \rangle^{\frac{1}{p}}$

**return** $\widehat{\mathcal{W}}_p + \Delta_{\hat{\mu}} + \Delta_{\hat{\nu}}$

---

Algorithm for bi-level quantization-based lower bound described in Section 3.5, using interpolation for upscaling the optimal coarse dual potentials, and c-transform to achieve optimized admissible dual potentials pair.

**Algorithm 5** Upscaling Lower Bound

**Require:** $\boldsymbol{\mu}, \boldsymbol{\nu} \in \Sigma_{N^d}$ on $\mathcal{X} = [N]^d$, $p \geq 1$, $\kappa \in \mathbb{N}$ and metric $\rho$.

$\quad \tilde{\mathcal{X}} \leftarrow \texttt{AvgPool}(\mathcal{X}; \kappa)$

$\quad \tilde{\boldsymbol{\mu}} \leftarrow \texttt{SumPool}(\boldsymbol{\mu}; \kappa)$

$\quad \tilde{\boldsymbol{\nu}} \leftarrow \texttt{SumPool}(\boldsymbol{\nu}; \kappa)$

$\quad \tilde{C} \leftarrow \{\rho(\tilde{x}_k, \tilde{x}_\ell)^p\}_{k\ell}$

$\quad \text{Solve } (\hat{\tilde{\boldsymbol{f}}}^*, \tilde{\boldsymbol{g}}^*) \leftarrow \underset{(\tilde{\boldsymbol{f}}, \tilde{\boldsymbol{g}}) \in \mathcal{R}(\tilde{C})}{\arg\max} \langle \tilde{\boldsymbol{f}}, \tilde{\boldsymbol{\mu}} \rangle + \langle \tilde{\boldsymbol{g}}, \tilde{\boldsymbol{\nu}} \rangle$

$\quad \hat{\boldsymbol{f}} \leftarrow \{R_{\tilde{\boldsymbol{f}}, \tilde{\mathcal{X}}}(x_i)\}_{i \in [N^d]}$

$\quad \boldsymbol{g} \leftarrow \left\{\min_i \rho(x_i, x_j)^p - \hat{f}_i\right\}_j$

$\quad \boldsymbol{f} \leftarrow \{\min_j \rho(x_i, x_j)^p - g_j\}_i$

$\quad \textbf{return } \left(\langle \boldsymbol{f}, \boldsymbol{\mu} \rangle + \langle \boldsymbol{g}, \boldsymbol{\nu} \rangle\right)^{\frac{1}{p}}$

## APPENDIX B   ADDITIONAL EXPERIMENTS

### APPENDIX B.1   DOTMARK

In this section we present additional figures and results evaluating the proposed Wasserstein bounds on the discrete optimal transport benchmark (DOTMark) (Schrieber et al., 2017) presented in the main text. The computational speed up of the proposed methods compared to the exact OT solver are summarized in Table 3. The results show that the quantization methods achieve the most significant speed ups. Notably, the dual upscaling method at $\kappa = 4$ are calculated in 0.2-2.2% of the time, while making almost no sacrifice in accuracy. Maintaining no more than 2.4% average error.

The relative accuracy of the proposed methods exponentially improves for large values of the exact Wasserstein distance as evident in Figures 4 and 5. Negative-valued lower bounds are trivially clipped to 0, when evaluate in the benchmark.

### APPENDIX B.2   EMDB

The Electron Microscopy Data Bank (EMDB) (wwPDB Consortium, 2024) is a repository of volumetric density maps that contains many interesting molecules that were reconstructed from cryogenic electron microscopy (cryo-EM) experiments. These reconstructions are estimates of the 3D electric potential at every point in the molecule. For our 3D experiments, we downloaded and processed four maps of famous molecules, detailed in Table 5 using the ASPIRE package (Wright et al., 2025). In Figure 2 you can see 3D renderings of these molecules that we generated using UCSF ChimeraX (Meng et al., 2023). The volumetric maps were downloaded from EMDB, masked inside a spherical region of radius 128 pixels, rotated around the Z axis in increments of 20 degrees and downscaled to $16 \times 16 \times 16$ voxels. The computational speed up is summarized in Table 6, showing that even at $\kappa = 2$ the quantization methods provide substantial speedups. Figure 7 shows the Wasserstein metrics and bounds between the 3D density map of the molecule in its base orientation and its rotations around the Z axis. The exact Wasserstein metric is shown as the thick black line with upper and lower bounds next to it using the various markers.

The quantization-based methods dominate in accuracy for the lower bounds of both $p \in \{1, 2\}$, whereas for the upper bounds of the Wasserstein-1 metric, the upper bound based on entropic regularization with $\varepsilon = 0.001N^p$ achieves the best accuracy, although at a significant computational cost. The triangular symmetry of EMDB-14621 (SARS-CoV-2 spike protein) and EMDB-2484 (Trimeric HIV-1 envelope glycoprotein) seen in Figure 7 are easily detectable as the dips at $120°$ rotation angle.

| | | Upper Bounds | | | | | | Lower Bounds | | | | | |
|---|---|---|---|---|---|---|---|---|---|---|---|---|---|
| | | Weighted-Cost | | Primal Upscaling | | Entropic Regularization | | Dual Upscaling | | Min-Cost | | Entropic Regularization | |
| | | $\kappa_2$ | $\kappa_4$ | $\kappa_2$ | $\kappa_4$ | $\varepsilon_1$ | $\varepsilon_4$ | $\kappa_2$ | $\kappa_4$ | $\kappa_2$ | $\kappa_4$ | $\varepsilon_1$ | $\varepsilon_4$ |
| Class | p | | | | | | | | | | | | |
| Classic Images | 1 | 4.4% $\pm 2.1\%$ | **0.3%** $\pm 0.2\%$ | 5.0% $\pm 2.6\%$ | **0.3%** $\pm 0.4\%$ | 19.8% $\pm 10.1\%$ | 16.0% $\pm 11.2\%$ | 4.9% $\pm 2.6\%$ | **0.2%** $\pm 0.1\%$ | 4.7% $\pm 2.3\%$ | 0.3% $\pm 0.1\%$ | 17.6% $\pm 7.8\%$ | 18.7% $\pm 13.3\%$ |
| | 2 | 6.1% $\pm 3.1\%$ | 0.4% $\pm 0.1\%$ | 6.1% $\pm 3.0\%$ | **0.3%** $\pm 0.1\%$ | 7.9% $\pm 6.6\%$ | 1.3% $\pm 0.5\%$ | 6.1% $\pm 3.0\%$ | **0.3%** $\pm 0.1\%$ | 6.1% $\pm 2.9\%$ | **0.3%** $\pm 0.1\%$ | 12.3% $\pm 10.4\%$ | 1.3% $\pm 0.6\%$ |
| Micro-scopy | 1 | 16.7% $\pm 5.6\%$ | **2.3%** $\pm 1.4\%$ | 20.7% $\pm 9.6\%$ | 6.6% $\pm 10.6\%$ | 128.0% $\pm 68.3\%$ | 90.7% $\pm 79.2\%$ | 14.4% $\pm 5.0\%$ | 2.2% $\pm 1.2\%$ | 15.7% $\pm 4.8\%$ | **1.9%** $\pm 1.0\%$ | 120.8% $\pm 57.2\%$ | 86.9% $\pm 81.3\%$ |
| | 2 | 15.4% $\pm 5.2\%$ | **1.9%** $\pm 1.2\%$ | 17.4% $\pm 5.9\%$ | 3.0% $\pm 3.4\%$ | 75.6% $\pm 51.3\%$ | 7.7% $\pm 4.9\%$ | 15.6% $\pm 5.1\%$ | 2.0% $\pm 1.3\%$ | 14.9% $\pm 5.0\%$ | **1.7%** $\pm 1.0\%$ | 99.9% $\pm 72.8\%$ | 8.1% $\pm 5.5\%$ |
| Shapes | 1 | 10.2% $\pm 3.6\%$ | **1.9%** $\pm 1.7\%$ | 15.1% $\pm 8.9\%$ | 10.8% $\pm 11.9\%$ | 172.9% $\pm 99.0\%$ | 68.8% $\pm 72.7\%$ | 7.8% $\pm 2.0\%$ | 1.5% $\pm 0.7\%$ | 8.5% $\pm 2.4\%$ | **1.4%** $\pm 1.2\%$ | 163.6% $\pm 88.8\%$ | 77.0% $\pm 77.6\%$ |
| | 2 | 7.9% $\pm 3.8\%$ | **1.5%** $\pm 1.8\%$ | 9.6% $\pm 5.6\%$ | 4.2% $\pm 6.4\%$ | 39.5% $\pm 45.0\%$ | 6.0% $\pm 11.6\%$ | 7.4% $\pm 2.6\%$ | 1.2% $\pm 0.6\%$ | 7.3% $\pm 3.0\%$ | **1.0%** $\pm 1.4\%$ | 56.2% $\pm 69.1\%$ | 5.7% $\pm 11.8\%$ |

Table 3: Computational time comparison of different methods showing the mean $\pm$ standard deviation of the relative computation time compared to exact OT solver. Each method is evaluated at different fidelity level $\kappa_2 = 2$ and $\kappa_4 = 4$ and different values of $\varepsilon_1 = 1 \cdot 10^{-3} N^p$ and $\varepsilon_4 = 4 \cdot 10^{-3} N^p$.

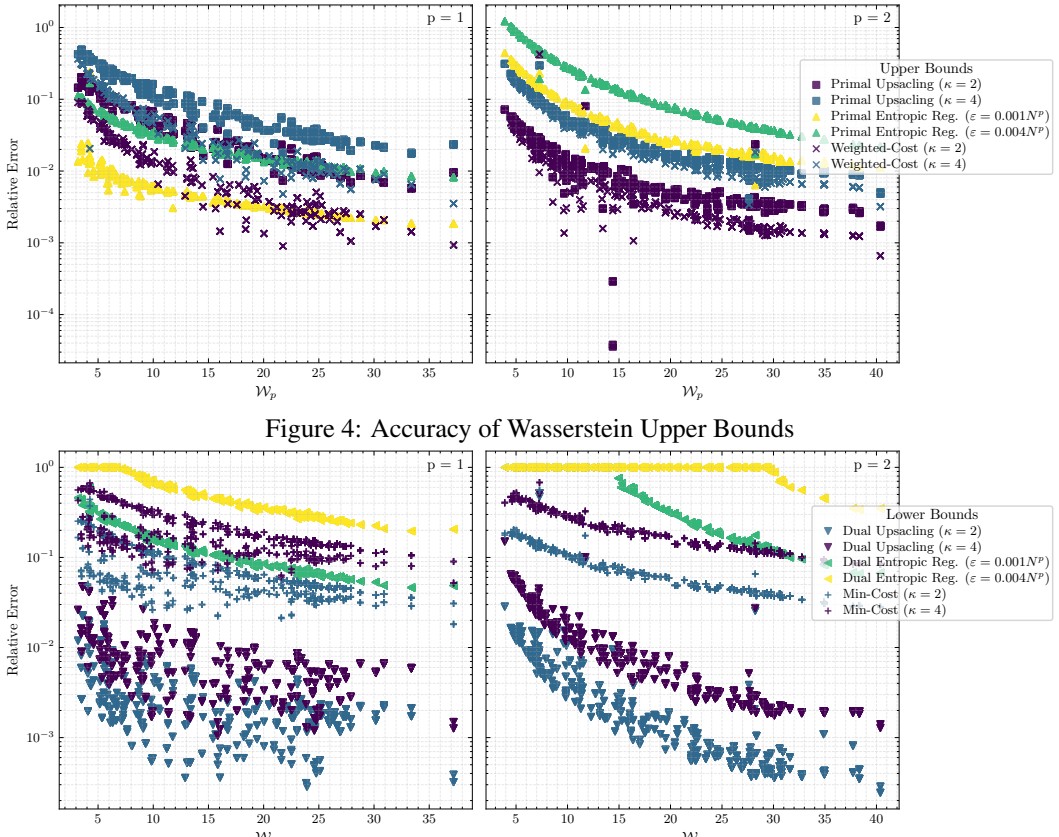

Figure 4: Accuracy of Wasserstein Upper Bounds

Figure 5: Accuracy of Wasserstein Lower Bounds. Negative-valued bounds are clipped to 0, evaluating as 100% relative error.

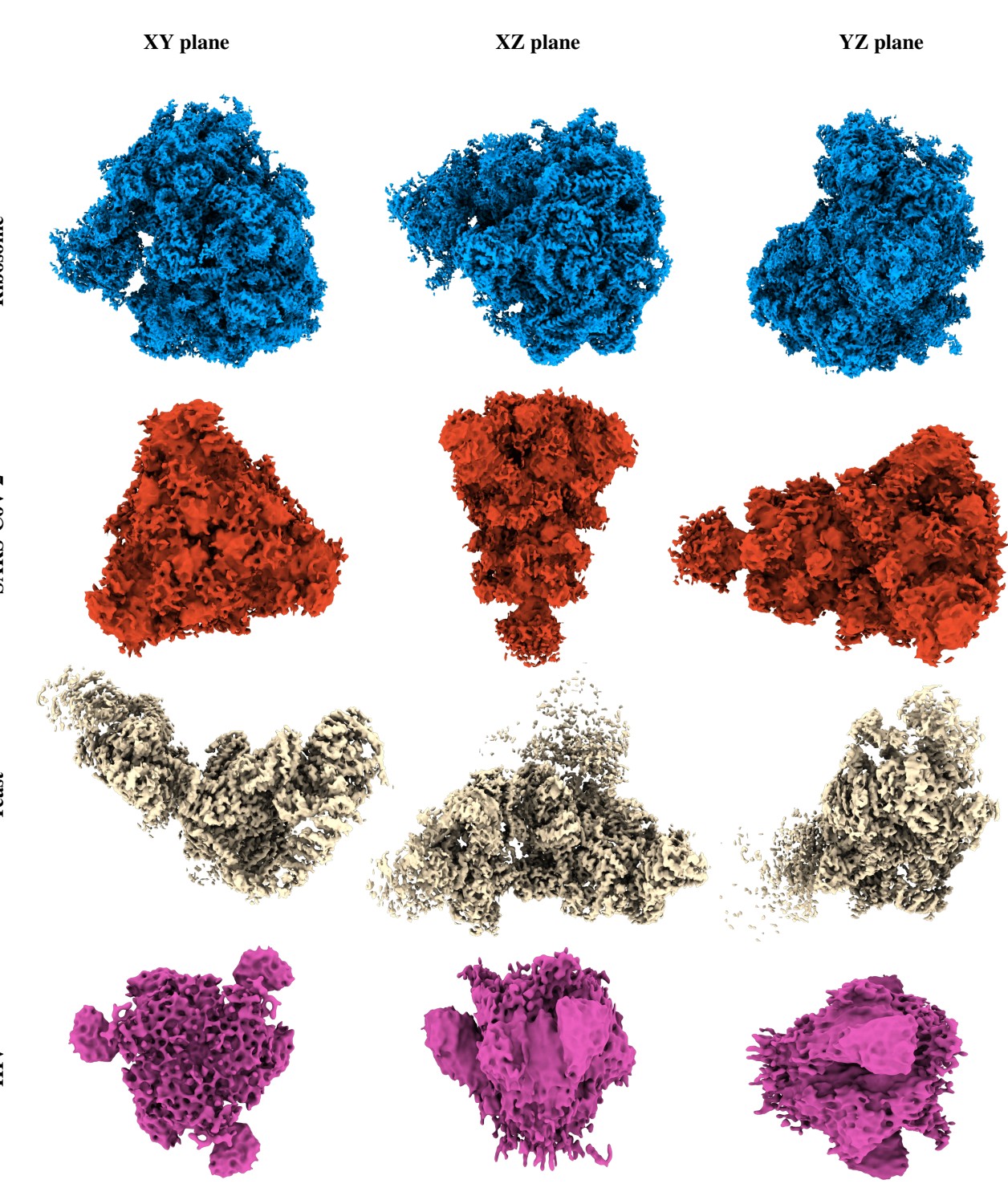

Figure 6: Isosurfaces of 3D molecular densities from the Electron Microscopy Data Bank (EMDB). In our experiments, the molecules on the left are rotated around the Z axis, which corresponds to the depth direction here. The middle and right columns show the same molecules rotated by 90 degrees around the X and Y axis (respectively).

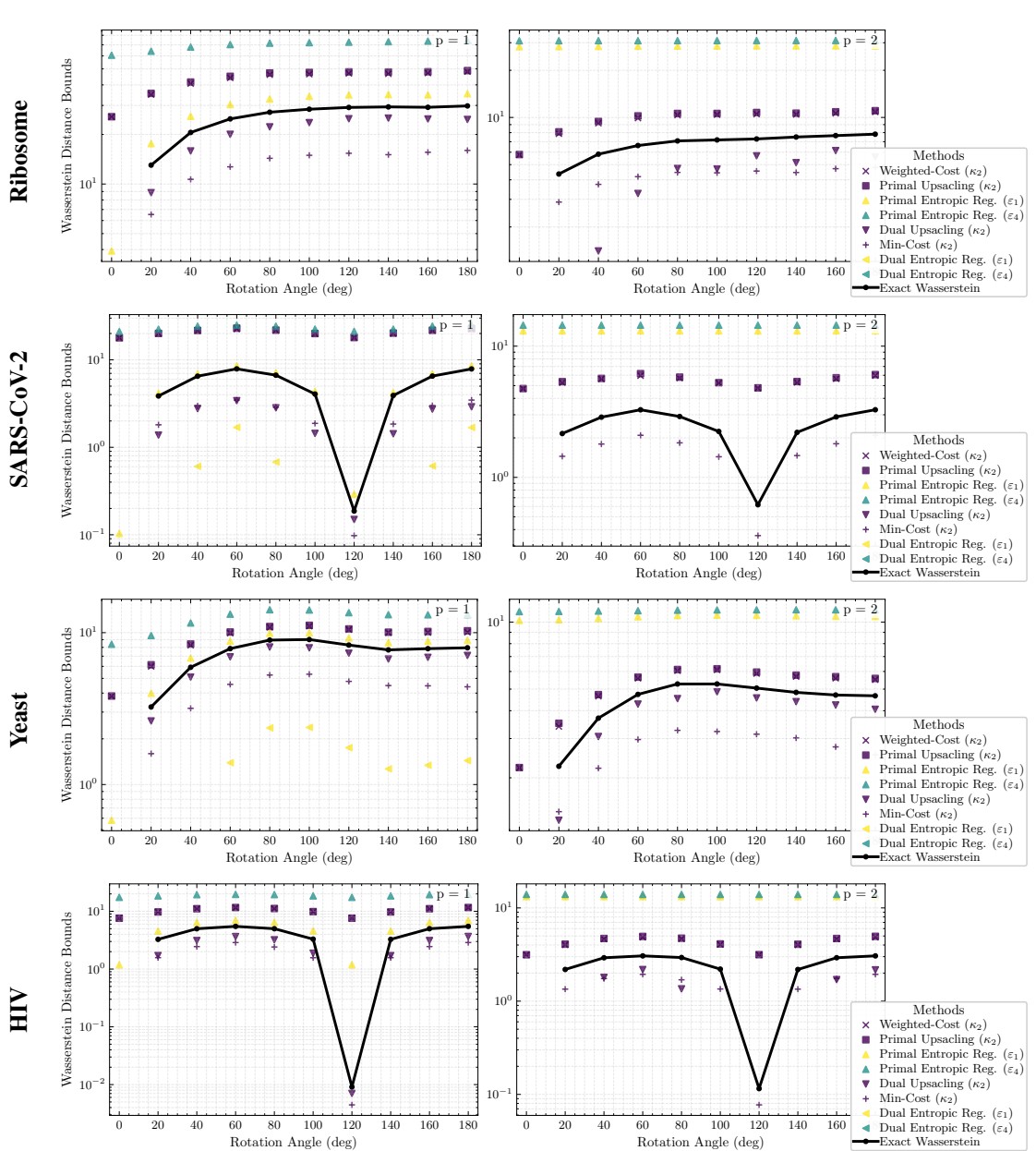

Figure 7: Wasserstein-$p$ metric and bounds between rotated 3D density maps of the molecules described in Table 5. From top to bottom: **Ribosome**, **SARS-CoV-2**, **Yeast**, **HIV**. The thick black line is the exact Wasserstein metric between the molecule and its rotated self, as a function of the rotation angle. The various upper and lower bounds are shown as different color markers. Note the drop around 120 degrees for the SARS-CoV-2 and HIV-1 spikes due to their 3-fold symmetry.

| | | Upper Bounds | | | | | | Lower Bounds | | | | | |
|---|---|---|---|---|---|---|---|---|---|---|---|---|---|
| | | Weighted-Cost | | Primal Upscaling | | Entropic Regularization | | Dual Upscaling | | Min-Cost | | Entropic Regularization | |
| Class | p | $\kappa_2$ | $\kappa_4$ | $\kappa_2$ | $\kappa_4$ | $\varepsilon_1$ | $\varepsilon_4$ | $\kappa_2$ | $\kappa_4$ | $\kappa_2$ | $\kappa_4$ | $\varepsilon_1$ | $\varepsilon_4$ |
| Classic Images | 1 | 3.1% | 11.0% | 9.6% | 23.0% | **0.9%** | 5.2% | **0.3%** | 0.7% | 10.0% | 27.0% | 24.0% | 88.0% |
| | | $\pm$ 2.0% | $\pm$ 7.1% | $\pm$ 4.1% | $\pm$ 9.9% | $\pm$ 0.5% | $\pm$ 2.1% | $\pm$ 0.2% | $\pm$ 0.4% | $\pm$ 6.4% | $\pm$ 16.0% | $\pm$ 8.3% | $\pm$ 15.0% |
| | 2 | **1.6%** | 7.9% | 2.2% | 8.8% | 14.0% | 44.0% | **0.7%** | 2.4% | 13.0% | 33.0% | 98.0% | 100.0% |
| | | $\pm$ 1.2% | $\pm$ 5.2% | $\pm$ 1.4% | $\pm$ 5.5% | $\pm$ 8.8% | $\pm$ 25.0% | $\pm$ 0.5% | $\pm$ 1.6% | $\pm$ 3.7% | $\pm$ 8.8% | $\pm$ 8.4% | $\pm$ 0.0% |
| Microscopy | 1 | 0.9% | 3.4% | 2.4% | 6.5% | **0.4%** | 2.0% | **0.4%** | 0.9% | 6.2% | 17.0% | 9.6% | 38.0% |
| | | $\pm$ 1.7% | $\pm$ 5.9% | $\pm$ 3.2% | $\pm$ 8.4% | $\pm$ 0.3% | $\pm$ 2.0% | $\pm$ 0.5% | $\pm$ 0.9% | $\pm$ 4.2% | $\pm$ 10.0% | $\pm$ 7.1% | $\pm$ 22.0% |
| | 2 | **0.5%** | 2.2% | 0.7% | 2.7% | 3.8% | 11.0% | **0.2%** | 0.7% | 5.5% | 16.0% | 30.0% | 90.0% |
| | | $\pm$ 0.7% | $\pm$ 3.5% | $\pm$ 1.0% | $\pm$ 3.9% | $\pm$ 5.9% | $\pm$ 18.0% | $\pm$ 0.4% | $\pm$ 1.2% | $\pm$ 3.4% | $\pm$ 8.4% | $\pm$ 32.0% | $\pm$ 19.0% |
| Shapes | 1 | 1.1% | 3.6% | 3.2% | 7.8% | **0.7%** | 2.6% | **0.5%** | 1.0% | 7.3% | 20.0% | 13.0% | 51.0% |
| | | $\pm$ 2.2% | $\pm$ 4.8% | $\pm$ 2.6% | $\pm$ 6.3% | $\pm$ 2.5% | $\pm$ 2.2% | $\pm$ 2.8% | $\pm$ 3.0% | $\pm$ 5.2% | $\pm$ 11.0% | $\pm$ 8.1% | $\pm$ 22.0% |
| | 2 | **1.2%** | 3.2% | 1.4% | 3.6% | 5.1% | 15.0% | **0.9%** | 1.7% | 7.7% | 20.0% | 52.0% | 99.0% |
| | | $\pm$ 4.5% | $\pm$ 5.0% | $\pm$ 4.5% | $\pm$ 5.0% | $\pm$ 6.2% | $\pm$ 17.0% | $\pm$ 5.1% | $\pm$ 5.6% | $\pm$ 6.1% | $\pm$ 9.2% | $\pm$ 34.0% | $\pm$ 9.6% |

Table 4: Accuracy comparison of different methods showing the mean $\pm$ standard deviation of the relative error computed across all the pairwise distances in the DOTmark class at $128 \times 128$ resolution. Each method is evaluated at different fidelity level $\kappa_2 = 2$ and $\kappa_4 = 4$ and different values of $\varepsilon_1 = 1 \cdot 10^{-3} N^p$ and $\varepsilon_4 = 4 \cdot 10^{-3} N^p$.

Table 5: Selected cryo-EM structures from the Electron Microscopy Data Bank (EMDB).

| Name | EMDB ID | Description |
|---|---|---|
| Ribosome | EMD-2660 | Ribosome of the Plasmodium falciparum parasite which causes malaria in humans (Wong et al., 2014) |
| SARS-CoV-2 | EMD-14621 | SARS-CoV-2 spike protein (Stagnoli et al., 2022) |
| Yeast | EMD-8012 | Yeast spliceosome (Nguyen et al., 2016) |
| HIV | EMD-2484 | HIV-1 trimeric spike pre-fusion (Bartesaghi et al., 2013) |

Table 6: Relative computation time compared to exact OT solver of $16 \times 16 \times 16$ downscaled EMDB density maps. Results show mean and standard deviation across rotation angles. Lower is better.

| Molecule | $p$ | Upper Bounds | | | | Lower Bounds | | | |
|---|---|---|---|---|---|---|---|---|---|
| | | Weighted-Cost $\kappa_2$ | Primal Upscaling $\kappa_2$ | Entropic Regularization $\varepsilon_1$ | $\varepsilon_4$ | Dual Upscaling $\kappa_2$ | Min-Cost $\kappa_2$ | Entropic Regularization $\varepsilon_1$ | $\varepsilon_4$ |
| Ribosome | 1 | **3.7%** ±3.2% | 175.6% ±56.2% | 156.2% ±111.3% | 155.8% ±111.1% | 74.5% ±21.8% | **3.5%** ±3.0% | 154.1% ±109.8% | 154.1% ±109.7% |
| | 2 | **12.4%** ±3.7% | 178.1% ±226.5% | 569.8% ±252.8% | 570.2% ±252.3% | **12.2%** ±3.6% | 12.5% ±5.0% | 563.2% ±249.8% | 563.6% ±249.6% |
| SARS-CoV-2 | 1 | **11.4%** ±26.7% | 164.6% ±96.4% | 69.1% ±4.7% | 62.7% ±15.1% | 69.9% ±35.7% | **2.2%** ±0.3% | 61.4% ±14.0% | 61.2% ±14.8% |
| | 2 | **19.7%** ±3.5% | 350.5% ±421.2% | 850.9% ±895.4% | 658.8% ±303.0% | 77.9% ±184.2% | **19.0%** ±5.7% | 642.5% ±295.5% | 641.6% ±294.0% |
| Yeast | 1 | 4.8% ±4.9% | 295.4% ±379.0% | 20.8% ±31.2% | **3.1%** ±3.5% | 127.5% ±153.4% | 3.3% ±3.3% | 5.6% ±4.6% | **0.9%** ±0.9% |
| | 2 | 12.7% ±0.7% | 172.2% ±69.3% | 919.4% ±497.3% | 828.9% ±215.6% | 13.3% ±1.0% | **11.6%** ±0.7% | 820.4% ±213.8% | 819.7% ±212.5% |
| HIV | 1 | **10.4%** ±23.6% | 243.1% ±215.3% | 163.1% ±127.3% | 163.0% ±127.1% | 99.6% ±89.8% | **2.9%** ±2.7% | 161.1% ±125.8% | 161.1% ±125.8% |
| | 2 | **40.7%** ±97.7% | 198.2% ±255.3% | 603.9% ±252.2% | 603.9% ±252.0% | 45.6% ±114.3% | **40.4%** ±99.0% | 597.0% ±249.3% | 597.0% ±249.3% |

## APPENDIX C   PROOFS OF THE MAIN CLAIMS

### APPENDIX C.1   WEIGHTED TOTAL VARIATION CORRECTION TERMS

*Proof of Lemma 3.2.*  Using the triangle inequality for the Wasserstein metric, we can write

$$\mathcal{W}_p(\mu, \nu) \leq \mathcal{W}_p(\mu, \hat{\mu}) + \mathcal{W}_p(\hat{\mu}, \hat{\nu}) + \mathcal{W}_p(\hat{\nu}, \nu) \tag{28}$$

controlling for each element separately, we have

$$\mathcal{W}_p(\mu, \hat{\mu}) \leq \mathcal{TV}_p^{\boldsymbol{w}}(\hat{\mu}, \mu) \qquad \text{and} \qquad \mathcal{W}_p(\hat{\nu}, \nu) \leq \mathcal{TV}_p^{\boldsymbol{w}}(\nu, \hat{\nu}) \tag{29}$$

by the property of weighted total variation, and

$$\mathcal{W}_p(\hat{\mu}, \hat{\nu}) = \left( \min_{\pi \in \Pi(\hat{\mu}, \hat{\nu})} \langle \pi, C \rangle \right)^{\frac{1}{p}} \leq \langle \hat{\pi}, C \rangle^{\frac{1}{p}} \tag{30}$$

by evaluating the transport cost using a coupling in the problem's original space.   □

*Proof of Proposition 3.3.*  Consider the definition of weighted total variation,

$$\mathcal{TV}_p(\hat{\mu}, \mu; w) + \mathcal{TV}_p(\nu, \hat{\nu}; w) \tag{31}$$

$$= 2^{1-\frac{1}{p}} \langle w, |\hat{\mu} - \mu| \rangle^{\frac{1}{p}} + 2^{1-\frac{1}{p}} \langle w, |\nu - \hat{\nu}| \rangle^{\frac{1}{p}}$$

$$= 2^{1-\frac{1}{p}} \left( \left( \sum \rho(\bar{x}, x)^p |\hat{\mu}_x - \mu_x| \right)^{\frac{1}{p}} + \left( \sum \rho(\bar{x}, x)^p |\nu_x - \hat{\nu}_x| \right)^{\frac{1}{p}} \right)$$

$$\leq 2^{1-\frac{1}{p}} \left( r \|\hat{\mu} - \mu\|_1^{\frac{1}{p}} + r \|\nu - \hat{\nu}\|_1^{\frac{1}{p}} \right) \qquad\qquad \text{bounding radius}$$

$$\leq 2^{2-\frac{2}{p}} \left( \|\hat{\mu} - \mu\|_1 + \|\nu - \hat{\nu}\|_1 \right)^{\frac{1}{p}} r \qquad\qquad \text{Jensen's inequality}$$

$$< 2^{2-\frac{2}{p}} \xi^{\frac{1}{p}} r \qquad\qquad\qquad\qquad \text{convergence criteria}$$

□

### APPENDIX C.2   WEIGHTED-COST UPPER BOUND

First, let us consider the following lemma discussing a coupling constructed ad hoc using coarsened measures.

**Lemma Appendix C.1.** *Let $\mu, \nu$ measures with set of admissible couplings $\Pi(\boldsymbol{\mu}, \boldsymbol{\nu})$, the trivial coupling $\pi_{\otimes}$, and $\tilde{\mu}$ and $\tilde{\nu}$ the respective coarsened measures. For any coupling $\tilde{\pi} \in \Pi(\tilde{\mu}, \tilde{\nu})$, the measure on the product space $\mathcal{X} \times \mathcal{Y}$*

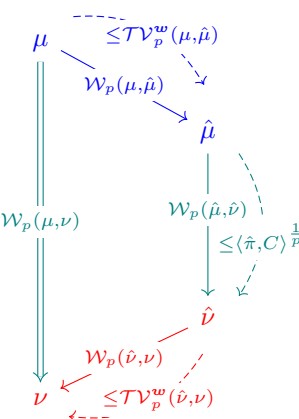

Figure 8: Diagram showing the relationship between measures and their Wasserstein distances, and the quantities bounding each element.

$$\pi_{\tilde{\pi}}(x,y) := \frac{\tilde{\pi}_{k_{\mathbb{X}}(x)\ell_{\mathbb{Y}}(y)}}{\mu(X_{k_{\mathbb{X}}(x)})\nu(Y_{\ell_{\mathbb{Y}}(y)})}\pi_{\otimes}(x,y) \tag{32}$$

*is an admissible coupling $\pi_{\tilde{\pi}} \in \Pi(\boldsymbol{\mu}, \boldsymbol{\nu})$, where the coarsening inverse index functions $k_{\mathbb{X}}(x)$, $\ell_{\mathbb{Y}}(y)$ are defined as $k_{\mathbb{X}}(x) := \{k : x \in X_k\}, \ell_{\mathbb{Y}}(y) := \{\ell : y \in Y_\ell\}$.*

*Proof of Lemma Appendix C.1.* Following Definition 1.1 (Coupling) from (Villani, 2009), one can show $\pi_{\tilde{\pi}}(x,y)$ Equation (32) is admissible. For $\varphi, \psi$ be any integrable measurable functions on $\mathcal{X}, \mathcal{Y}$ respectively, than $\pi_{\tilde{\pi}}(x,y)$ admits

$$
\begin{aligned}
&\int_{\mathcal{X}\times\mathcal{Y}} \big(\varphi(x) + \psi(y)\big)\mathrm{d}\pi_{\tilde{\pi}}(x,y) \\
&= \sum_{k,\ell} \int_{X_k \times Y_\ell} \big(\varphi(x) + \psi(y)\big)\mathrm{d}\pi_{\tilde{\pi}}(x,y) \\
&= \sum_{k,\ell} \int_{X_k \times Y_\ell} \big(\varphi(x) + \psi(y)\big)\frac{\boldsymbol{\Pi}_{k\ell}}{\mu(X_k)\nu(Y_\ell)}\mathrm{d}\mu(x)\mathrm{d}\nu(y) \qquad \text{plug-in coupling's definition} \\
&= \sum_{k,\ell} \Big(\frac{\tilde{\pi}_{k\ell}}{\mu(X_k)\nu(Y_\ell)}\int_{X_k \times Y_\ell}\varphi(x)\mathrm{d}\mu(x)\mathrm{d}\nu(y) + \frac{\tilde{\pi}_{k\ell}}{\mu(X_k)\nu(Y_\ell)}\int_{X_k \times Y_\ell}\psi(y)\mathrm{d}\mu(x)\mathrm{d}\nu(y)\Big) \\
&= \sum_{k,\ell}\frac{\tilde{\pi}_{k\ell}}{\mu(X_k)}\int_{X_k}\varphi(x)\mathrm{d}\mu(x) + \sum_{k,\ell}\frac{\tilde{\pi}_{k\ell}}{\nu(Y_\ell)}\int_{Y_\ell}\psi(y)\mathrm{d}\nu(y) \qquad \text{sum over marginals} \\
&= \sum_k \int_{X_k}\varphi(x)\mathrm{d}\mu(x) + \sum_\ell \int_{Y_\ell}\psi(y)\mathrm{d}\nu(y) \\
&= \int_{\mathcal{X}}\varphi(x)\mathrm{d}\mu(x) + \int_{\mathcal{Y}}\psi(y)\mathrm{d}\nu(y)
\end{aligned}
\tag{33}
$$

$\square$

Next, we consider the transport cost of such a coupling.

**Lemma Appendix C.2.** *The transport loss assigned by the cost $c(x,y)$ and a coupling $\pi_{\tilde{\pi}}$ identifies with the coarse transport loss assigned by marginally weighted cost $\bar{C}$ Equation (15) and the coarse coupling $\tilde{\pi}$,*

$$\langle \pi_{\tilde{\pi}}, C \rangle = \langle \tilde{\pi}, \bar{C} \rangle \tag{34}$$

*Proof.*

$$
\begin{aligned}
\langle \pi_{\tilde{\pi}}, C \rangle &= \sum_{i,j} c(x_i, y_j)\pi_{\tilde{\pi}}(x_i, y_j) = \sum_{k,\ell}\sum_{\substack{x \in X_k \\ y \in Y_\ell}} c(x,y)\pi_{\tilde{\pi}}(x,y) \\
&= \sum_{k,\ell}\sum_{\substack{x \in X_k \\ y \in Y_\ell}} c(x,y)\frac{\tilde{\pi}_{k\ell}}{\mu(X_k)\nu(Y_\ell)}\pi_{\otimes}(x,y) \\
&= \sum_{k,\ell}\frac{1}{\mu(X_k)\nu(Y_\ell)}\sum_{\substack{x \in X_k \\ y \in Y_\ell}} c(x,y)\mu(x)\nu(y)\,\tilde{\pi}_{k\ell} \\
&= \sum_{k,\ell} \bar{C}_{k\ell}\tilde{\pi}_{k\ell} = \langle \tilde{\pi}, \bar{C} \rangle
\end{aligned}
\tag{35}
$$

$\square$

Finally, we can write

*Proof of Theorem 3.4.* Based on admissibility of $\pi_{\tilde{\pi}}$ shown in Lemma Appendix C.1 the transport cost

$$\langle \pi_{\tilde{\pi}}, C \rangle \geq L_C(\mu, \nu), \ \forall \tilde{\pi} \in \Pi(\tilde{\mu}, \tilde{\nu}). \tag{36}$$

In particular, for $\tilde{\pi}^* = \arg\min_{\tilde{\pi} \in \Pi(\tilde{\mu}, \tilde{\nu})} \langle \tilde{\pi}, \bar{C} \rangle$,

$$\langle \tilde{\pi}^*, \bar{C} \rangle = \langle \pi_{\tilde{\pi}^*}, C \rangle \geq L_C(\mu, \nu) \tag{37}$$

by the identity shown in Lemma Appendix C.2. $\square$

APPENDIX C.3   MIN-COST LOWER BOUND

*Proof of Theorem 3.5.* Consider

$$\pi^* = \underset{\pi \in \Pi(\boldsymbol{\mu}, \boldsymbol{\nu})}{\arg\min} \langle \pi, C \rangle \tag{38}$$

and coarsening of the optimal coupling

$$\hat{\pi}^*_{k\ell} := \sum_{\substack{x \in X_k \\ y \in Y_\ell}} \pi^*(x, y) \tag{39}$$

such that,

$$
\begin{aligned}
L_C(\boldsymbol{\mu}, \boldsymbol{\nu}) &= \langle \pi^*, C \rangle \tag{40} \\
&= \sum_{\substack{x \in \mathcal{X} \\ y \in \mathcal{Y}}} \rho(x, y)^p \pi^*(x, y) \\
&= \sum_{k, \ell} \sum_{\substack{x \in X_k \\ y \in Y_\ell}} \rho(x, y)^p \pi^*(x, y) \\
&\geq \sum_{k, \ell} C^{\min}_{k\ell} \sum_{\substack{x \in X_k \\ y \in Y_\ell}} \pi^*(x, y) \\
&= \langle \hat{\pi}^*, C^{\min} \rangle \\
&\geq \min_{\tilde{\pi} \in \Pi(\tilde{\boldsymbol{\mu}}, \tilde{\boldsymbol{\nu}})} \langle \tilde{\pi}, C^{\min} \rangle \\
&= L_{C^{\min}}(\tilde{\mu}, \tilde{\nu}).
\end{aligned}
$$

$\square$

APPENDIX C.4   UP-SCALED COUPLING

*Proof of Lemma 3.6.* Let $\tilde{\pi}^* \in \Pi(\tilde{\boldsymbol{\mu}}, \tilde{\boldsymbol{\nu}})$ be the coarse optimal coupling and $\mathbf{K}$ be the positive valued normalized kernel tensor satisfying $\sum_{t \in [\kappa]^{2d}} \mathbf{K}_t = 1$. Recall that $\hat{\mathbf{P}} = \tilde{\mathbf{P}}^* \otimes \mathbf{K}$ and $\hat{\pi}$ is obtained by reshaping $\hat{\mathbf{P}}$.

First, we show that the sum of all elements equals 1:

$$
\begin{aligned}
\sum_{i=1}^{N^d} \sum_{j=1}^{N^d} \hat{\pi}_{ij} &= \sum_{t \in [\kappa]^{2d}} \sum_{k=1}^{n^d} \sum_{\ell=1}^{n^d} \tilde{\pi}^*_{k\ell} \mathbf{K}_t \tag{41} \\
&= \sum_{k=1}^{n^d} \sum_{\ell=1}^{n^d} \tilde{\pi}^*_{k\ell} \sum_{t \in [\kappa]^{2d}} \mathbf{K}_t \\
&= \sum_{k=1}^{n^d} \sum_{\ell=1}^{n^d} \tilde{\pi}^*_{k\ell} \cdot 1 = 1
\end{aligned}
$$

where the last equality follows from $\tilde{\pi}^*$ being a coupling.

Second, we show that $\hat{\pi}$ is non-negative. Since $\tilde{\pi}^*$ is a coupling, it is non-negative, and $\mathbf{K}$ is a positive-valued kernel, their tensor product $\hat{\mathbf{P}}$ and its reshaped form $\hat{\pi}$ are non-negative.

Thus, $\hat{\pi}$ satisfies all the properties of a coupling measure. $\qquad\square$

## APPENDIX D   APPROXIMATION ERROR ANALYSIS

In this section, we derive the theoretical tightness rates for the proposed upper and lower bounds. Let us define covering Radius ($r_\kappa$) and cell Diameter ($\delta_\kappa$), the maximum distance from any point in the cell to its center and the maximum distance between any two points within the cell, respectively, for a coarse cell $X_k$ with center $\tilde{x}_k$:

$$r_\kappa = \sup_{x \in X_k} \|x - \tilde{x}_k\|_2 = \frac{1}{2}\kappa\sqrt{d} \qquad \delta_\kappa = \sup_{x,x' \in X_k} \|x - x'\|_2 = \kappa\sqrt{d} \tag{42}$$

We assume the underlying ground metric $\rho(x, y)$ is $L_\rho$-Lipschitz with respect to the Euclidean norm. Furthermore, for the transport cost $c(x, y) = \rho(x, y)^p$ on bounded domain $\mathcal{X} \times \mathcal{X}$ with diameter $D$, for $p \geq 1$ the cost $c(x, y)$ is locally Lipschitz with constant $L_c = pD^{p-1}L_\rho$.

### APPENDIX D.1   WEIGHTED-COST UPPER BOUND

This bound approximates the transport solving the optimal transport using the expected cost between pair of coarse cells over the trivial coupling.

**Lemma Appendix D.1** (Cost Variation). *Let $c(x, y)$ be $L_c$-Lipschitz. For any two coarse cells $X_k$ and $Y_\ell$, and any specific points $x \in X_k, y \in Y_\ell$, the deviation of the weighted average cost $\bar{C}_{k\ell}$ from the specific cost $c(x, y)$ is bounded by the sum of the cell diameters:*

$$|\bar{C}_{k\ell} - c(x, y)| \leq 2L_c\delta_\kappa = 2L_c\kappa\sqrt{d}. \tag{43}$$

*Proof.* By the triangle inequality and Lipschitz continuity, for any $u \in X_k, v \in Y_\ell$:

$$|c(u, v) - c(x, y)| \leq L_c(\rho(u, x) + \rho(v, y)) \leq L_c(\delta_\kappa + \delta_\kappa).$$

Since $\bar{C}_{k\ell} = \mathbb{E}_{u \sim \mu|_{X_k}, v \sim \nu|_{Y_\ell}}[c(u, v)]$, the expectation is bounded by the maximum deviation. $\qquad\square$

**Theorem Appendix D.2.** *The approximation error of the Weighted-Cost Upper Bound satisfies:*

$$\overline{\mathcal{W}}_p^\otimes(\mu, \nu) - \mathcal{W}_p(\mu, \nu) \leq 2D^{p-1}L_\rho \cdot \kappa\sqrt{d}.$$

*Proof.* Let $\pi^*$ be the optimal coupling for the fine-scale problem. We construct a coarse coupling $\tilde{\pi}^*$ by aggregating $\pi^*$ such that $\tilde{\pi}^*_{k\ell} = \sum_{x \in X_k, y \in Y_\ell} \pi^*(x, y)$. The bound value $\overline{L} = L_{\bar{C}}(\tilde{\mu}, \tilde{\nu})$ minimizes the weighted-cost over coarse couplings, thus $\overline{L} \leq \langle \tilde{\pi}^*, \bar{C} \rangle$. The gap between $\overline{L}$ and $L_C(\mu, \nu)$ is bounded by:

$$Gap \leq \sum_{k,\ell} \tilde{\pi}^*_{k\ell}\bar{C}_{k\ell} - \sum_{x,y} \pi^*(x, y)c(x, y) = \sum_{k,\ell} \sum_{x \in X_k, y \in Y_\ell} \pi^*(x, y)(\bar{C}_{k\ell} - c(x, y)).$$

Applying Lemma Appendix D.1, the term in parentheses is bounded by $2L_c\kappa\sqrt{d}$. Summing over the probability mass yields the result, assuming $\overline{L} \geq L_C \geq 1$:

$$\overline{\mathcal{W}}_p^\otimes - \mathcal{W}_p = \overline{L}_C^{1/p} - L_C^{1/p} \tag{44}$$

$$\leq \frac{1}{p}(\overline{L}_C - L_C) \tag{45}$$

$$\leq \frac{1}{p}(L_c\kappa\sqrt{d}). \tag{46}$$

$\qquad\square$

.

**Theorem Appendix D.3.** *The approximation error of the Min-Cost Lower Bound satisfies:*

$$\mathcal{W}_p(\mu, \nu) - \underline{\mathcal{W}}_p^{\min}(\mu, \nu) \leq 2D^{p-1}L_\rho \cdot \kappa\sqrt{d}.$$

*Proof.* Using the same aggregated coupling $\tilde{\pi}^*$ as above, the lower bound value is at most $\langle \tilde{\pi}^*, C^{\min} \rangle$. The error is bounded by the maximum deviation between the actual fine cost $c(x, y)$ and the optimistic coarse cost $C_{k\ell}^{\min} = \min_{u \in X_k, v \in Y_\ell} c(u, v)$. Let $(u^*, v^*)$ be the minimizers for the cell pair. For any $x \in X_k, y \in Y_\ell$:

$$c(x, y) - c(u^*, v^*) \leq L_c(\rho(x, u^*) + \rho(y, v^*)) \leq 2L_c\delta_\kappa.$$

This holds for all transported mass, yielding a total error of $2L_c\kappa\sqrt{d}$. $\qquad\square$

**Theorem Appendix D.4.** *Neglecting the total variation correction terms (assuming $\xi \to 0$), the error of the Primal Upscaling Upper Bound satisfies:*

$$\overline{\mathcal{W}}_p(\mu, \nu) - \mathcal{W}_p(\mu, \nu) \leq L_\rho \cdot \kappa\sqrt{d}.$$

*Proof.* The primal upscaling method constructs a solution based on the optimal coarse plan. Its cost is equivalent to $\mathcal{W}_p(\tilde{\mu}, \tilde{\nu})$ where $\tilde{\mu}, \tilde{\nu}$ are the coarsened measures supported on cell centers. By the triangle inequality of the Wasserstein metric:

$$\mathcal{W}_p(\mu, \nu) \leq \mathcal{W}_p(\tilde{\mu}, \tilde{\nu}) + \mathcal{W}_p(\mu, \tilde{\mu}) + \mathcal{W}_p(\nu, \tilde{\nu}).$$

The term $\mathcal{W}_p(\mu, \tilde{\mu})$ represents the quantization error of moving mass from the fine grid $x \in X_k$ to the center $\tilde{x}_k$. The displacement is bounded by the covering radius $r_\kappa$.

$$\mathcal{W}_p(\mu, \tilde{\mu}) \leq L_\rho r_\kappa = L_\rho \frac{\kappa}{2}\sqrt{d}.$$

Summing the quantization errors for $\mu$ and $\nu$ yields $L_\rho\kappa\sqrt{d}$. $\qquad\square$

**Theorem Appendix D.5.** *Let the optimal dual potential $f$ be $L_c$-Lipschitz. Assuming the optimal transport cost is bounded away from zero such that $L_C(\mu, \nu) \geq 1$, the approximation error of the Wasserstein distance satisfies:*

$$\mathcal{W}_p(\mu, \nu) - \underline{\mathcal{W}}_p(\mu, \nu) \leq 2D^{p-1}L_\rho \cdot \kappa\sqrt{d}.$$

*Proof.* Let $L_C(\mu, \nu) = \mathcal{W}_p^p(\mu, \nu)$ be the exact optimal transport cost, and $\underline{L}_C = \underline{\mathcal{W}}_p^p(\mu, \nu)$ be the approximate lower bound cost.

First, we bound the error in the cost domain. The error is driven by the interpolation of the dual potentials. Let $f^*$ be the optimal potential on the fine grid. $f^*$ is $c$-concave and thus inherits the Lipschitz constant $L_c$ from the cost function (Villani, 2009). The method approximates $f^*$ using $\hat{f}$, a multilinear interpolation of the coarse grid values. For an $L_c$-Lipschitz function, the sup-norm error of linear interpolation on a cell of diameter $\delta_\kappa$ is bounded by the variation of the function over the cell:

$$\|f^*(x) - \hat{f}(x)\|_\infty \leq L_c \cdot \delta_\kappa.$$

Since the dual objective is linear in the potentials for each measure, the gap is bounded by $L_C - \underline{L}_C \leq 2L_c\delta_\kappa = 2L_c\kappa\sqrt{d}$. $\qquad\square$

# APPENDIX E  TABLE OF NOTATIONS

Table 7: Table of Notations

| Notation | Category | Description |
|---|---|---|
| *Sets and Spaces* | | |
| $\mathbb{R}_+$ | Set | Non-negative real numbers |
| $[n]$ | Set | Set of integers $\{1, \ldots, n\}$ |
| $\Sigma_N$ | Space | Probability simplex $\{(p_1, \ldots, p_N) \in \mathbb{R}_+^N : \sum_i p_i = 1\}$ |
| $\mathcal{X}, \mathcal{Y}$ | Set | Point sets where measures are defined |
| $\mathbb{X}$ | Set | Set of non-overlapping hypercubes covering the grid |
| $\tilde{\mathcal{X}}, \tilde{\mathcal{Y}}$ | Set | Coarse grids (set of hypercube centers) |
| $X_k, Y_\ell$ | Set | Individual hypercubes in the partition |
| *Measures and Vectors* | | |
| $\mathbf{0}_n, \mathbf{1}_n$ | Vector | All-zeros and all-ones vectors in $\mathbb{R}^n$, respectively |
| $\mu, \nu$ | Measure | Discrete probability measures |
| $\boldsymbol{\mu}, \boldsymbol{\nu}$ | Vector | Vector representations of measures $\mu, \nu$ |
| $\tilde{\mu}, \tilde{\nu}$ | Measure | Coarsened measures |
| $\tilde{\boldsymbol{\mu}}, \tilde{\boldsymbol{\nu}}$ | Vector | Vector representations of coarsened measures |
| $\boldsymbol{f}, \boldsymbol{g}$ | Vector | Kantorovich potentials |
| $\boldsymbol{a}, \boldsymbol{b}$ | Vector | Sinkhorn scaling vectors |
| *Matrices and Tensors* | | |
| $C$ | Matrix | Ground-cost matrix |
| $\tilde{C}$ | Matrix | Coarse cost matrix (center-based) |
| $\bar{C}$ | Matrix | Coarse cost matrix (average-based) |
| $\pi$ | Matrix | Transport plan (coupling matrix) |
| $\pi^*$ | Matrix | Optimal transport coupling |
| $\tilde{\pi}$ | Matrix | Coarse coupling |
| $\mathbf{K}$ | Tensor | Normalized kernel tensor |
| *Functions and Operations* | | |
| $\rho$ | Function | Distance metric |
| $\mathcal{W}_p$ | Function | Wasserstein-$p$ metric |
| $L_C$ | Function | Optimal transport cost for ground-cost $C$ |
| $\mathcal{TV}_p^{\boldsymbol{w}}$ | Function | Weighted total variation |
| $\otimes$ | Operation | Tensor product |
| $\odot, \oslash$ | Operation | Pointwise multiplication, division |
| $\langle \cdot, \cdot \rangle$ | Operation | Standard vector/matrix inner product |
| *Parameters and Constants* | | |
| $p$ | Scalar | Order of Wasserstein metric ($p \geq 1$) |
| $\kappa$ | Scalar | Scale factor for coarsening |
| $\xi$ | Scalar | Convergence threshold for fitting |
| $N$ | Scalar | Side length of regular grid |
| $d$ | Scalar | Dimension of the space |
| $n$ | Scalar | Side length of coarse grid ($n = N/\kappa$) |
| $\#S$ | Scalar | Cardinality (size) of set $S$ |
| $r$ | Scalar | Radius of space $\mathcal{X}$ |
| $\Delta_{\hat{\mu}}, \Delta_{\hat{\nu}}$ | Scalar | Marginal weighed total variation corrections |
| *Code* | | |
| `AvgPool`, `SumPool` | Function | Average/sum pooling layer with identical kernel size and stride |
| `mean` | Function | Mean of a set of points |
| `reshape` | Function | Cardinality preserving tensor shape transformation |

