# OpenReview forum: "Quantization bounds for Wasserstein metrics"
_ICLR.cc/2026/Conference — Submitted to ICLR 2026_

### Official Review · Reviewer_rPMX · 2025-10-30

**Soundness:** 3
**Presentation:** 2
**Contribution:** 3
**Rating:** 6
**Confidence:** 3

**Summary:**

The paper concerns discrete probability measures on uniform grids. The authors develop efficient algorithms for bounding Wasserstein distance $\mathcal{W}_p$ (e.g., earth mover's distance) between such measures. The search for methods that compute exact upper and lower bounds (rather than just approximations) is motivated by the fact that such bounds enable the use of branch-and-bound, A* path finding and other search techniques for optimization problems involving $\mathcal{W}_p$. The proposed methods work as follows. First, a coarser grid is defined, and measures are downscaled. Second, a special cost matrix for the coarse grid is constructed (possibly depending on original measures), and the downscaled Kantorovich problem is solved exactly. Finally, the solution is upscaled and adjusted to obtain a valid bound. In numerical experiments, bounds based on the Sinkhorn algorithm are used as a baseline. The proposed methods outperform the baseline in these experiments both in terms of speed and accuracy.

**Strengths:**

1. The paper contributes to a relatively underexplored yet valuable area of research.
2. It presents theorems establishing the validity of the proposed bounds, along with a complexity analysis.
3. The theoretical results are supported by experimental evaluation.
4. The paper is relatively easy to follow.

**Weaknesses:**

1. Overview of existing bounds is lacking, see questions
2. Quality of the presentation should be improved
3. Numerical comparison to other bounds (apart from Sinkhorn-based) could also be added

**Questions:**

1. Could you please mention existing bounds in the "Related work" section? For example, Table 1 in the paper *Fast Dataset Search with Earth Mover’s Distance* by W. Yang et al. provides references to several bounds for the earth mover’s distance that would be worth citing and, where applicable, comparing against in the experiments.
2. In Figure 2, "Dual Entropic Reg." bounds are present in the legend but not on the plots. Am I missing something?
3. In formula (7), what do you mean by $|X_i|$? Also, please define SumPool and AvgPool more clearly as a function of 2 arguments (as you use it subsequently). Moreover, please define it for both usual measures and couplings.
4. In formula (8), you use entropy. Please define it explicitly.
5. Lines 259-260: please define a *normalized* kernel.
6. You write both "upscaling" and "up-scaling". I suggest that you settle on the first option, and do same for "downscaling".

Addressing the concerns is important for keeping (or possibly raising) the score.

Typos and minor mistakes:
- Line 25: "at 2D/3D" should be changed to "in 2D/3D".
- Line 135: "section 3.1" should be changed to "Section 3.1".
- Line 175: it seems that the period between "not" and "since" was placed by mistake.
- Line 195: "Proofs for 3.2" should be changed to "Proofs for Lemma 3.2 and Proposition 3.3".
- Lines 260-261: did you mean to place ":" instead of the period before the formula for $\hat{\mathbf{P}}$?
- Line 269: it seems that the period between "factors" and "yielding" was placed by mistake.
- Line 277: "than" should be removed.
- Line 282: please turn the part that starts with "Negligible" into a proper sentence.
- Line 306: "Since by" sounds unnatural.
- Figure 2: change "Upsacling" to "Upscaling" in the legend (4 times).

---

> ### Author Response · Authors · 2025-12-01
> **Rebuttal by Authors**
>
> We wish to thank the reviewer for putting the time to thoroughly read the paper and for providing a high-quality review with specific points for improvement. This important service to the community is unfortunately becoming quite rare these days. We did our best to address all of the points you have raised and improved the manuscript thanks to your feedback.
> Here is a point-by-point response:
>
> "1. Could you please mention existing bounds in the "Related work" section? For example, Table 1 in the paper Fast Dataset Search with Earth Mover’s Distance by W. Yang et al. provides references to several bounds for the earth mover’s distance that would be worth citing and, where applicable, comparing against in the experiments."
>
> We have expanded our Related Work section to include these methods. They provide significantly looser approximations and for the different domain of sampled point-clouds, so we do not think that adding them as baselines to the experiments makes sense.
>
> "2. In Figure 2, "Dual Entropic Reg." bounds are present in the legend but not on the plots. Am I missing something?"
>
> The "Dual Entropic Reg." bounds are so bad in this case that they give a negative result. Since the plot is in log-scale they cannot be displayed. We have added a clarification in the revised version.
>
> "3. In formula (7), what do you mean by $|X_i|$ ? Also, please define SumPool and AvgPool more clearly as a function of 2 arguments (as you use it subsequently). Moreover, please define it for both usual measures and couplings."
>
> The notation $|X_i|$ means the cardinality of the set $X_i$. We've changed the notation to #$X_i$ and explained it in the text. We also added a line to the notation table.
>
> "4. In formula (8), you use entropy. Please define it explicitly."
>
> Done.
>
> "5. Lines 259-260: please define a normalized kernel."
>
> Done.
>
> "6. You write both "upscaling" and "up-scaling". I suggest that you settle on the first option, and do same for "downscaling"."
>
> Done.
>
> Additionally, all the typos and minor mistakes are fixed in the revised version. Thank you!

---

### Official Review · Reviewer_39aw · 2025-10-30

**Soundness:** 2
**Presentation:** 2
**Contribution:** 2
**Rating:** 4
**Confidence:** 3

**Summary:**

This paper proposes a family of quantization-based methods to efficiently approximate the p-Wasserstein distance between discrete probability measures defined on regular grids. The main goal is to produce upper and lower bounds on Wasserstein distance that are significantly faster to compute.

**Strengths:**

1, fast upper/lower bounds for Wasserstein distance on regular grids are provided.
2, Four complementary estimators (weighted-cost UB, min-cost LB, primal upscaling UB, dual upscaling LB) are introduced.

**Weaknesses:**

1, The proposed method primarily reduces computational cost by down-scaling the sample size through grid coarsening, rather than introducing a fundamentally new optimal-transport formulation or theoretical advance. While the quantization framework offers practical acceleration, it largely depends on existing solvers applied to smaller grids, with modest algorithmic novelty. As such, the broader impact on the optimal-transport or machine-learning community may be limited. The approach trades accuracy for efficiency in a relatively straightforward way and does not substantially deepen theoretical understanding or extend applicability to new data types.

2, The paper reports the average relative bounding region only empirically, with no theoretical justification or convergence analysis.

3, A more useful theory could be bounding the relative error (Upper Bound − Wasserstein distance)/Wasserstein distance, which directly measures overestimation of the true distance. This could be theoretically bounded under smoothness or regularity assumptions on the underlying distributions (e.g., Lipschitz or Holder continuity of the densities).

4, It’s unclear how to extend the bounds to irregular meshes or point clouds.

**Questions:**

1, Can you provide tightness rates for each bound as a function of the coarse factor, dimension , and simple distribution classes?

2, It would be nice to include a sensitivity analysis of how accuracy depends on grid resolution.

---

> ### Author Response · Authors · 2025-12-01
> **Rebuttal by Authors**
>
> "1, Can you provide tightness rates for each bound as a function of the coarse factor, dimension , and simple distribution classes?
>
> 2, It would be nice to include a sensitivity analysis of how accuracy depends on grid resolution."
>
> Thank you for this suggestion. The paper was updated to include tightness analysis, with proofs for the error bounds available in Appendix D. The approximation error of the suggested methods is governed of the quantization factor $\kappa$ and scales with $O(\kappa\sqrt{d})$.
>
> Additionally, since the error bound is additive, it is independent of the original grid resolution.

---

### Official Review · Reviewer_g7Zw · 2025-10-31

**Soundness:** 2
**Presentation:** 2
**Contribution:** 2
**Rating:** 2
**Confidence:** 3

**Summary:**

This paper introduces a family of grid-based techniques for computing upper and lower bounds on the Wasserstein distance between discrete distributions. The upper bounds are obtained by coarsening the input distributions onto a regular grid and solving an exact optimal transport problem using block-averaged costs or coarse-grid couplings that are then upscaled to the original resolution, with marginal mismatches corrected via weighted total variation. The constructions are designed to ensure that the resulting estimates always overestimate the true transport cost. The lower bounds are derived either from nearest-neighbor mappings, either by coarse OT with blockwise minimal costs or by interpolating duals refined by a c-transform to enforce dual feasibility.

The authors do not establish any approximation ratio or error bound quantifying how close their upper and lower bounds are to the exact value as a function of the grid resolution, dimension, or scaling factor $\kappa$. Consequently, the framework ensures correctness but offers no provable tightness guarantees, relying instead on empirical evidence to demonstrate that the bounds are often reasonably close in practice.

Furthermore, several existing methods already provide provable upper and lower bounds on the Wasserstein distance. For instance, quad-tree or hierarchical constructions yield an $O(\log ⁡n)$-approximation upper bound for $W_1$(see, e.g., Indyk & Thaper 2003; Backurs et al. 2022), while the nearest-neighbor mapping, also known as the Chamfer distance, serves as a simple and widely used lower bound. The paper does not directly compare its grid-based bounds to these established techniques. This omission makes it difficult to assess the theoretical significance of the proposed methods.

**Strengths:**

The claims appear correct, and the implementations are careful and well-executed. The experiments demonstrate meaningful computational speed-ups, and the writing is clear and well structured.

**Weaknesses:**

The paper lacks theoretical novelty and does not provide provable approximation guarantees. Existing hierarchical and Chamfer-based methods already achieve provable bounds, yet no direct comparison is made. The contribution is therefore primarily empirical, with limited new theoretical insight.

**Questions:**

Please address the concerns I have raised in my review.

---

> ### Author Response · Authors · 2025-12-01
> **Rebuttal by Authors**
>
> "The authors do not establish any approximation ratio or error bound quantifying how close their upper and lower bounds are to the exact value as a function of the grid resolution, dimension, or scaling factor $\kappa$"
>
> Thank you for pointing out this important omission in the original submission. The paper was updated to include an analysis of the approximation error of the bounds, with proofs available in Appendix D. For Lipschitz-continuous ground costs, the approximation error of all the scales like $O(\kappa\sqrt{d})$.
>
> "Furthermore, several existing methods already provide provable upper and lower bounds on the Wasserstein distance. For instance, quad-tree or hierarchical constructions yield an O(log n) approximation upper bound for (see, e.g., Indyk & Thaper 2003; Backurs et al. 2022), while the nearest-neighbor mapping, also known as the Chamfer distance, serves as a simple and widely used lower bound. The paper does not directly compare its grid-based bounds to these established techniques. This omission makes it difficult to assess the theoretical significance of the proposed methods"
>
> These papers, while influential in the field, are not relevant as a direct comparison to the methods discussed in our paper. The methods differ by one or more of:
> 1. Focusing on point-cloud setting, while we discuss methods for discrete measures on a 2D/3D grid.
> 2. Solving only for $W_1$, while our results apply for any $p  \ge 1$;
> 3. Yielding multiplicative error bound with $O(\log n)$ factor. These are much worse than our additive error bounded by $O(\kappa\sqrt{d})$ term our case.
>
> We have added references to these (and related) works in the revised Related Work section.

---

### Official Review · Reviewer_b4Ra · 2025-10-31

**Soundness:** 3
**Presentation:** 2
**Contribution:** 2
**Rating:** 4
**Confidence:** 2

**Summary:**

This paper introduces efficient methods for computing fast approximations that bound the Wasserstein metric between discrete distributions on a regular grid. Experiments on 2D and 3D data show the proposed approach achieves greater computational efficiency and accuracy than entropic optimal transport-based bounds.

**Strengths:**

1) This approach outperforms bounds based OT entropic methods, demonstrating less deviation compared to other methods.
2) The authors propose four tools for bounding from weighted-cost upper bound to dual upscaling lower bound
3) This finding allows developing new type of generative models based on OT

**Weaknesses:**

1) If I am not mistaken, the authors validated their method only for 2D and 3D dimensional setups. Unfortunately, there is no information about scalabity of the method in high-dimensional spaces (for example:100 or 1000, experiments even with Gaussian distributions seem enough).
2) see questions

**Questions:**

1) Is there any opportunity to continue this method in continuous space?
2) Could you provide more high-dimensional experiments with discrete Gaussian distributions?
3) Your method performs equally with independent transport plan and true transport plan, doesn't it?
4) How does the behaviour of convergence change depending on number of samples?

---

> ### Author Response · Authors · 2025-12-01
> **Rebuttal by Authors**
>
> Thank you for time and consideration put in to reviewing our paper, following is a point by point response to the raised questions.
>
> "1. Is there any opportunity to continue this method in continuous space?"
>
> 1. Yes! in principle, our methods can be made to work in any space that can be subdivided/clustered in some way, with additional technical requirements depending on the methods. For example, the min-cost lower bound would require compactness of the subregions (otherwise the minimum is not guaranteed) and the weighted-cost upper-bound would require the convergence of the integrals of $\rho(x,y)d\mu(x) d\nu(y)$ on each subregion. The theorems in the paper discuss discrete measures on regular grids for conciseness and readability, and also because these settings are driven by motivating applications in image and volumetric processing.  We think that extending the work to broader domains is a very interesting direction for future research and have updated the paper to include this point.
>
> "2. Could you provide more high-dimensional experiments with discrete Gaussian distributions?"
>
> 2. While, as we show in paper, the efficiency gains of our bounds are exponential in the dimension. However, since the methods involve solving the exact linear-program for the quantized measures on a grid, they are not practical for high-dimensional grids, due to both memory and runtime complexity. Since the number of variables in the LP scale like $(N/\kappa)^d$ where $N$ is the grid side-length, $\kappa$ is the coarsening factor and $d$ is the dimension. We do not think this is an actual limitation, since we are not aware of any applications that compute Wasserstein metrics on grids beyond 3D.
>
> "3. Your method performs equally with independent transport plan and true transport plan, doesn't it?"
>
> 3. It's not clear to us what you mean by independent and true transport plan. The paper proposes a few different methods, each taking a different approach for manipulating the transport plans.
>
> "4. How does the behavior of convergence change depending on number of samples?"
>
> 4. Our paper does not discuss sampling at all but deals only with signals on a grid. e.g. it  associates an $n$ by $n$ image with a discrete measure on the $n \times n$ grid. However, the approximation error of the suggested methods depends on the quantization factor $\kappa$ as $O(\kappa\sqrt{d})$. We added an Appendix D in the revised manuscript to include this analysis.

---

### Meta-Review · Area_Chair_cTje · 2025-12-28

**Summary:**

This paper proposes a family of quantization-based methods for computing the upper and lower bounds of p-Wasserstein metric on regular grids. The idea of upper bounds is to coarsen input distribution onto regular grids, solve the exact OT problem and then upscaled to the initial resolution, while ensuring estimates are above the true transport cost. The idea of lower bounds is derived from nearest-neighbor mappings. The paper shows better computational efficiency and accuracy than entropic OT-based bounds on 2D and 3D data.

The reviewers raised several common weakness and questions:

- lack of provable tightness of upper and lower bounds, and convergence analysis;
- insufficient comparison with existing relevant methods for $W_p$ distance approximation;
- extension to irregular meshes, point clouds, and higher dimensions.

**Reviewer Concerns:**

In response, the authors revised the paper to include a simple tightness bound in terms of quantization factor and dimension. More references were added to the Related Work section.

Some of the major concerns are not addressed in the rebuttal. It remains unclear how the approximation and tightness depend on the grid structure, and whether a relative error (a more direct measure) can be derived from the current theory.

**Reviewer Scores:**

4 reviewers submitted their comments and scores (4/2/4/6) with confidence (3/3/2/3), with average score 4 and average confidence 2.75.

---

### Decision · Program_Chairs · 2026-01-26

Reject